Resource

# Deciphering the regulatory landscape of fetal and adult γδ T-cell development at single-cell resolution

Sagar[1], Maria Pokrovskii[2], Josip S Herman[1,3,4], Shruti Naik[5], Elisabeth Sock[6], Patrice Zeis[1,3,4], Ute Lausch[7], Michael Wegner[6] (ID), Yakup Tanriver[7,8], Dan R Littman[2,9] & Dominic Grün[1,10,*] (ID)

## Abstract

γδ T cells with distinct properties develop in the embryonic and adult thymus and have been identified as critical players in a broad range of infections, antitumor surveillance, autoimmune diseases, and tissue homeostasis. Despite their potential value for immunotherapy, differentiation of γδ T cells in the thymus is incompletely understood. Here, we establish a high-resolution map of γδ T-cell differentiation from the fetal and adult thymus using single-cell RNA sequencing. We reveal novel sub-types of immature and mature γδ T cells and identify an unpolarized thymic population which is expanded in the blood and lymph nodes. Our detailed comparative analysis reveals remarkable similarities between the gene networks active during fetal and adult γδ T-cell differentiation. By performing a combined single-cell analysis of *Sox13, Maf,* and *Rorc* knockout mice, we demonstrate sequential activation of these factors during IL-17-producing γδ T-cell (γδT17) differentiation. These findings substantially expand our understanding of γδ T-cell ontogeny in fetal and adult life. Our experimental and computational strategy provides a blueprint for comparing immune cell differentiation across developmental stages.

**Keywords** double negative progenitors; fetal and adult thymus; gamma delta T cells; gamma delta T-cell differentiation; single-cell RNA sequencing
**Subject Category** Immunology
The EMBO Journal (2020) 39: e104159

## Introduction

Advances in single-cell technologies have enabled an unbiased classification of immune cell types and the inference of their differentiation trajectories on a genome-wide scale. However, efforts to reconstruct high-resolution differentiation trajectories have mainly focused on the hematopoietic compartment in the bone marrow (Paul *et al*, 2015; Nestorowa *et al*, 2016; Olsson *et al*, 2016; Velten *et al*, 2017; Tusi *et al*, 2018). In particular, comparisons of immune cell developmental processes across distinct periods of life —such as fetal and adult stages—are still lacking. A number of immune cell types first develops in embryonic tissues and continues their development in the adult counterparts. For example, T cells first arise in the fetal thymus and continue developing in the adult thymus. The thymus supports the development of two T-cell lineages—αβ and γδ T cells. While a significant amount of research has focused on understanding the development of αβ T cells, the γδ lineage remains understudied. This lineage represents a unique developmental paradigm in immune cell ontogeny blurring the demarcation of the innate and adaptive arms of immunity (Lanier, 2013). γδ T cells exert innate-like rapid immune responses by recognizing a broad spectrum of molecules including non-peptide antigens through TCR-dependent and TCR-independent mechanisms (Hayday, 2009). They are the earliest T cells to develop in the embryonic thymus, and their differentiation and effector functions are developmentally pre-programmed: Rearrangement of defined T-cell receptor (TCR) γ chains occurs at discrete time points and is followed by selective migration to individual epithelial tissues such as skin, lung, intestine, and reproductive tract. γδ T-cell development continues after birth in the thymus albeit utilizing different TCR γ chains (Carding & Egan, 2002), and the naïve cells are believed to mature in the secondary lymphoid organs (Chien *et al*, 2013).

Although the role of γδ T cells in infection, tumors, and autoimmune diseases is widely recognized, their intrathymic differentiation is not well understood (Munoz-Ruiz *et al*, 2017). γδ T-cell differentiation is an attractive developmental model to explore whether similar pathways are used to direct the differentiation of distinct sub-types of the same lineage at different stages of life. Such

1 Max Planck Institute of Immunobiology and Epigenetics, Freiburg, Germany
2 Molecular Pathogenesis Program, The Kimmel Center for Biology and Medicine of the Skirball Institute, New York University School of Medicine, New York, NY, USA
3 Faculty of Biology, University of Freiburg, Freiburg, Germany
4 International Max Planck Research School for Molecular and Cellular Biology (IMPRS-MCB), Freiburg, Germany
5 Department of Pathology and Ronald O. Perelman Department of Dermatology, NYU School of Medicine, New York, NY, USA
6 Institut für Biochemie, Emil-Fischer-Zentrum, Friedrich-Alexander-Universität Erlangen-Nürnberg, Erlangen,Germany
7 Institute of Medical Microbiology and Hygiene, University Medical Center Freiburg, Freiburg, Germany
8 Department of Internal Medicine IV, University Medical Center Freiburg, Freiburg, Germany
9 The Howard Hughes Medical Institute, New York University School of Medicine, New York, NY, USA
10 CIBSS-Centre for Integrative Biological Signaling Studies, University of Freiburg, Freiburg, Germany
  *Corresponding auhtor. Tel: +49 7615 108490; E-mail: gruen@ie-freiburg.mpg.de

studies may also serve as a blueprint for comparing the differentiation of other immune cell lineages across developmental time points and tissues. In order to decipher the transcriptional landscape of γδ T-cell differentiation in the fetal and adult murine thymus at single-cell resolution, we utilized multi-color flow cytometry to enrich cell populations encompassing all differentiation stages of the γδ T-cell lineage and profiled them using single-cell RNA sequencing (scRNA-seq). We identified a number of novel sub-types, in particular, an unpolarized $Ccr9^+$ $S1pr1^+$ thymic population which was expanded in the peripheral blood and in lymph nodes and produces TNF-α, IFN-γ, and IL-2 upon stimulation. Further, we predicted continuous differentiation trajectories and inferred gene regulatory networks (GRNs) governing γδ T-cell differentiation. We performed a multi-layered comparative analysis of fetal and adult differentiation and observed remarkable similarities between fetal and adult gene modules activated during the process of T-cell commitment and γδ T-cell differentiation. A focused analysis revealed that *Sox13*, *Maf*, and *Rorc* act in a sequential manner to drive γδT17 differentiation in the fetal and adult thymus.

## Results

### scRNA-seq of T-cell progenitors and γδ T cells from the fetal and adult mouse thymus

To investigate and compare the transcriptional landscape of γδ T-cell differentiation during fetal and adult life, we isolated thymocyte subsets from fetal (embryonic day 17.5–18.5) and adult (6–7 weeks old) mice utilizing established cell surface markers (Fig EV1A and E). These populations comprise bipotent αβ/γδ T-cell precursors—c-KIT$^+$ double negative (DN) 1, DN2, and DN3 (Fig EV1B and F), CD25$^+$ γδ T-cell precursors (Fig EV1C and G), CD24$^+$ (immature) and CD24$^-$ (mature) γδ T-cell populations from fetal thymus (Fig EV1D), pan γδ T cells (mainly containing CD24$^+$ immature cells) and CD24$^-$ (mature) γδ T cells (Fig EV1H), and IFN-γ-producing CD122$^+$ γδ T cells from the adult thymus (Fig EV1I) (Shibata *et al*, 2008; Narayan *et al*, 2012). Using this strategy, we sampled the entire γδ T-cell differentiation trajectory at two developmental time points. Single cells were sorted into 384-well plates by flow cytometry to perform scRNA-seq according to our published mCEL-Seq2 protocol (Hashimshony *et al*, 2016; Herman *et al*, 2018) and analyzed with RaceID3 (Fig 1A). After removing low-quality cells based on low total transcript numbers, 4,146 and 3,235 cells were retained for analysis of fetal and adult data, respectively. We did not observe batch-associated variability and recovered all cell types across replicates. We identified 30 and 24 clusters in the fetal and adult dataset, respectively, demonstrating substantial heterogeneity within conventionally defined γδ T-cell populations (Fig 1B–G). This suggests the existence of previously unknown sub-types or sub-states (see Table EV1 for differentially expressed marker genes).

### Characterizing heterogeneity in the early double negative T-cell progenitors

We first characterized heterogeneity in the DN1-DN3 progenitors capable of giving rise to both αβ and γδ T-cell lineages. RaceID3

classified fetal c-KIT$^+$ DN1 cells, also known as early thymic progenitors (ETPs), into two distinct clusters (14 and 15; Fig 1B–D); cluster 15 comprises *Flt3*$^+$ cells and thus may represent the most naïve ETPs (Fig EV1J). Cluster 14 expresses higher levels of *Tcf7*, required for T-cell development, *Il2ra* (encoding CD25), a cell surface marker of DN2 and DN3 progenitors as well as TCR β and γ constant chains—*Trbc1*, *Trbc2*, *Tcrg-C1*, and *Tcrg-C2* (Fig EV2A) (Godfrey *et al*, 1993; Schilham *et al*, 1998). These data suggest that fetal ETPs can be divided into naïve progenitors and cells starting to express genes associated with T-cell commitment. Similarly, RaceID3 classified adult ETPs into two clusters (2 and 4; Fig 1E–G); cluster 4 exhibits higher expression of receptor tyrosine kinase *Kit*, while cells in cluster 2 upregulate genes associated with DNA replication and cell cycle progression, e.g., *Mcm2, Mcm5, Mcm6, Mki67,* and *Pcna* (Fig EV2D), suggesting that adult ETPs unlike their fetal counterparts exhibit cell cycle-associated heterogeneity. Consistently, gene set enrichment analysis (GSEA) of differentially expressed genes between fetal and adult ETPs revealed preferential expression of proliferation-associated genes at the fetal stage, while genes associated with death receptor, G protein-coupled receptor (GPCR), and Toll-like receptor (TLR) signaling pathways were upregulated at the adult stage (Fig EV2I).

T-cell commitment occurs at the DN2 stage, which is subdivided into the uncommitted DN2a and the committed DN2b state, a transition marked by downregulation of *Kit* and upregulation of the T-cell commitment factor *Bcl11b* (Yui *et al*, 2010; Kueh *et al*, 2016). We identified two major subsets of DN2 progenitors in the fetal and adult thymus—clusters 8 and 13 (fetus) and clusters 3 and 6 (adult; Fig 1B–G). Fetal DN2 cluster 8 shows an upregulation of *Bcl11b* while expressing ETP genes such as *Ifitm1* and *Pim1* (Fig EV2B). Cluster 13 shows higher expression of T-cell-related genes such as *Lck*, *Thy1*, *Cd3d*, *Cd3e*, and *Lat,* indicating commitment (Fig EV2B). We found similar results in the adult dataset: Cluster 3 exhibits an ETP-like gene expression signature (e.g., *Adgrg1*, *Adgrg3*), whereas cells in cluster 6 upregulate *Bcl11b* as well as *Cd3g*, *Cd3e*, *Cd3d*, *Lck*, and *Lat* (Fig EV2E). Therefore, our unbiased single-cell analysis recapitulates the sub-division of fetal and adult DN2 cells into DN2a- and DN2b-like subsets. Differential gene expression analysis revealed an upregulation of recombination-associated genes such as *Rag1*, *Rag2,* and pre-T-cell antigen receptor alpha *Ptcra* in fetal DN2 cells, whereas adult DN2 cells still expressed ETP-related markers such as *Adgrg1*, *Adgrg3,* and *Cpa3,* indicating that fetal T-cell progenitors start to activate the recombination machinery earlier than their adult counterparts (Fig EV2J).

In the T-cell committed DN3 compartment, fetal and adult cells are composed of two clusters each—clusters 5 and 11 (fetus) and clusters 11 and 12 (adult; Fig 1B–G). Fetal cluster 11 and adult cluster 12 express *Rag1*, *Rag2*, *Notch1,* and *Ptcra* (Fig EV2C and F). These clusters have minimal levels of cell cycle-related genes (Fig EV2G and H) and, hence, represent cells undergoing recombination. Fetal cluster 5 and adult cluster 11 comprise proliferating cells expressing *Mki67*, *Top2a*, *Lig1,* and *Pcna* and may represent post-selected DN3 cells (Fig EV2C and F). GSEA revealed that fetal DN3 cells are more proliferative than adult DN3 cells (Fig EV2K). Accordingly, adult DN3 subsets expressed gene sets associated with recombination and chromatin modification (Fig EV2F and K). Collectively, our analysis reveals different subsets of DNs in fetal and adult thymi and demonstrates a continuous transcriptional shift in early T-cell

progenitors undergoing commitment, recombination, and selection tightly coupled with proliferation during T-cell development.

### Temporal differences between fetal and adult early thymopoiesis

We next attempted to identify DN cells sharing a similar transcriptional program at fetal and adult stages. This can be formulated as a classification problem: Given a set of classifiers, i.e., cluster medoids in one dataset (e.g., fetal dataset having 30 clusters), the objective is to identify non-negative weights for each single cell in a given query (e.g., a single cell in the adult data) under the constraint that the weights sum up to one. We quantified these weights by mapping this to a quadratic optimization problem, which can be solved by quadratic programming. We plotted the weights assigned

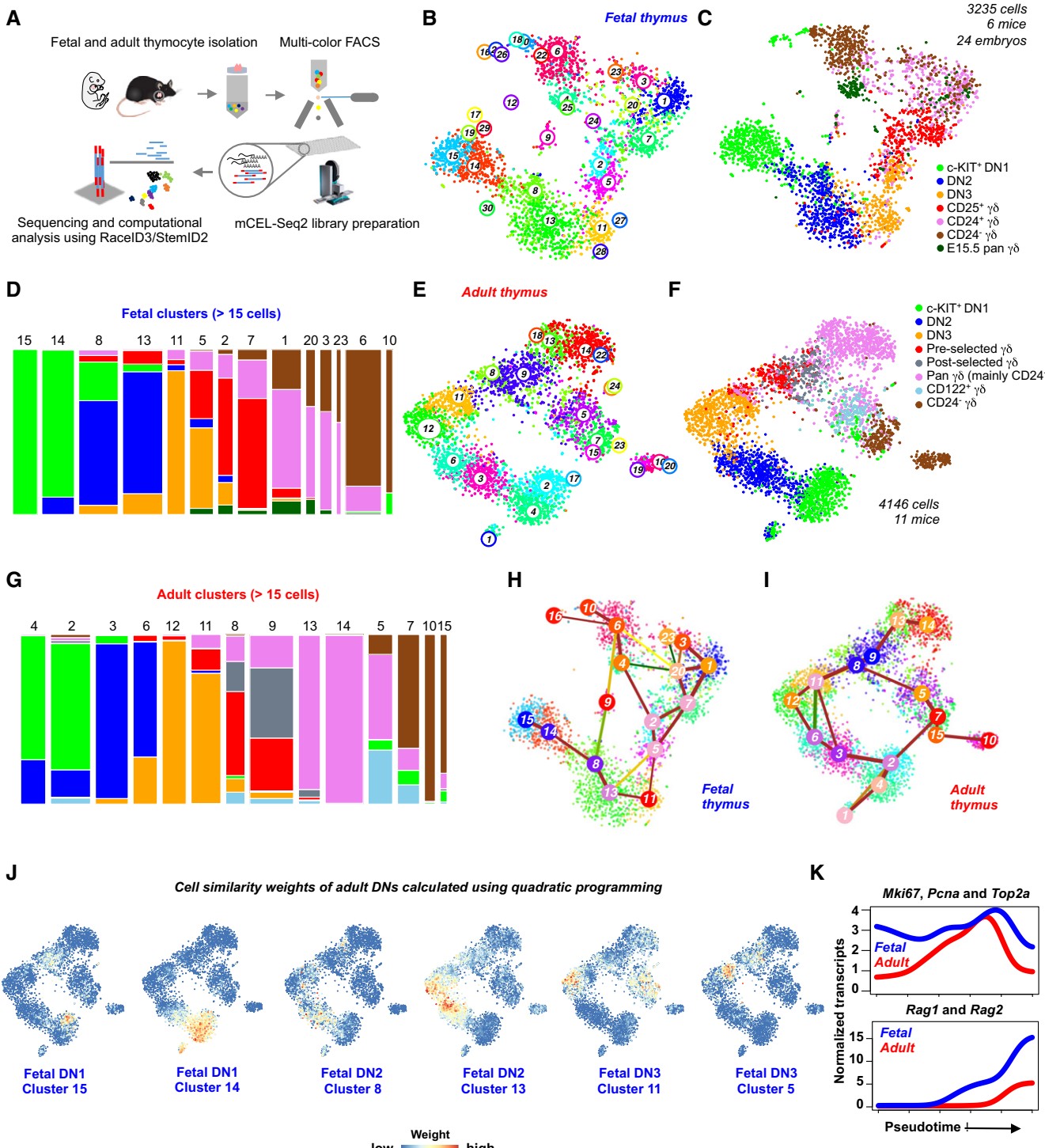

**Figure 1.**

◀

**Figure 1.   Single-cell RNA sequencing (scRNA-seq) of γδ T-cell differentiation from the fetal and adult thymus.**

A      Schematic representation of the workflow used for single-cell sorting, library preparation, and data analysis.

B, C    t-SNE representation based on transcriptome similarities showing 30 clusters identified by the RaceID3 algorithm (B) and the sorted cell populations (C) from fetal thymi. Colors represent different cell types sorted using fluorescence-activated cell sorting (FACS; $n$ = 2 minimum independent experiments for sorting each population, 24 mouse fetal thymi, embryonic day (E) 17.5-E18.5).

D      Bar plot showing the contribution of sorted cell types to the fetal clusters comprising more than 15 cells. The width of the bars is proportional to the cell numbers in the clusters.

E, F    t-SNE representation based on transcriptome similarities showing 24 clusters identified by the RaceID3 algorithm (E) and the sorted cell types (F) from the adult thymi ($n$ = 2 minimum independent experiments for sorting each cell type, 11 adult thymi, 6- to 7-week-old female mice).

G      Bar plot showing the contribution of sorted cell types to the adult clusters comprising more than 15 cells. The width of the bars is proportional to the cell numbers in the clusters.

H, I    Inferred lineage tree of fetal (H) and adult (I) γδ T-cell differentiation using the StemID2 algorithm. Only significant links are shown ($P < 0.01$). The color of the link indicates the $-\log_{10}P$. The color of the vertices indicates the transcriptome entropy. The thickness indicates the link score, reflecting how densely a link is covered with cells.

J      t-SNE representation of the adult thymus dataset showing the weights for fetal DN clusters calculated using quadratic programming. Color scale represents weights on the scale of 0–1.

K      Pseudo-temporal expression profiles of *Mki67*, *Pcna*, and *Top2a* (top) as well as *Rag1* and *Rag2* (bottom) along the DN1 to DN3 differentiation trajectories. The lines indicate the pseudo-temporal expression values derived by a local regression of expression values across the ordered cells. Blue and red lines indicate the fetal and adult data, respectively.

to all queries, i.e., all single cells in the adult dataset for each classifier, i.e., fetal cluster medoid, on the t-distributed stochastic neighbor embedding (t-SNE) representation and found that cells in adult cluster 2 had high weights for fetal cluster 15, indicating that these two clusters have the highest transcriptome similarity across the datasets (Fig 1J). Of note, cells in both of these clusters expressed higher levels of *Flt3*, suggesting that they represent the most naïve ETPs at the respective developmental time point. Similarly, the fetal ETP cluster 14 resembled adult cluster 4 (Fig 1J). DN2 and DN3 clusters revealed a remarkable correspondence between fetal and adult stages (Fig 1J). Interestingly, we found that cells in adult cluster 12, mainly composed of DN3 cells undergoing recombination, displayed higher weights for fetal cluster 13, which mainly contained DN2 cells, indicating that the genes associated with recombination are activated earlier during fetal T-cell development (Fig 1D, G and J). As a complementary approach, we inferred differentiation trajectories of fetal and adult thymopoiesis from the DN1 to the DN3 stage with the StemID2 algorithm (Grün *et al*, 2016; Herman *et al*, 2018) (Fig 1H and I). Pseudo-temporal gene expression profiles revealed that fetal DN progenitors overall express higher levels of cell cycle-related genes (e.g., *Mki67*, *Pcna,* and *Top2a*) and that recombination-related genes (e.g., *Rag1* and *Rag2)* are switched on already at the fetal DN2 stage (Fig 1K). Overall, we demonstrate a shift in temporal dynamics of these pathways during fetal and adult thymopoiesis.

**Identifying transcriptionally heterogeneous immature and mature γδ T-cell subsets**

In the fetal and adult thymus, the earliest CD25$^+$ γδ T-cell progenitors are mainly proliferating (i.e., express *Mki67* and *Top2a*) and specifically express *Cd5*, a gene implicated in regulating TCR signaling (Fig 2C and F) (Azzam *et al*, 1998, 2001). CD25$^+$ progenitors from the fetal and adult thymus were classified into two clusters—clusters 2 and 7 (fetus) and clusters 8 and 9 (adult; Fig 2A and D). All clusters expressed genes such as *Cd5*, *Cd28*, *Hivep3,* and *Lef1* at different levels (Fig 2C and F). Differential gene expression analysis between fetal clusters 2 and 7 revealed higher expression of γδT17-associated genes (e.g., *Sox13* and *Blk*) in cluster 7, indicating that these cells are already primed toward the γδT17 fate. In contrast,

cells in cluster 2 express higher levels of TCR signaling genes such as *Nfatc1* and *Prkch*, and thus may experience stronger TCR signals (Fig 2C). Similarly, adult cluster 9 cells express elevated levels of γδT17-associated genes such as *Sox13* and *Blk*, while cluster 8 upregulate *Nfatc1* and *Prkch* (Fig 2F). Consistently, quadratic programming confirmed a correspondence between fetal cluster 2 and adult cluster 8, which upregulate TCR signal strength-related genes, and between γδT17-primed adult cluster 9 and fetal cluster 7 (Fig EV3A). In conclusion, our results are in accordance with the TCR signal strength-based effector differentiation model (Zarin *et al*, 2015).

We next characterized the immature CD24$^+$ and the mature CD24$^-$ compartments in the fetal thymus (Fig 2A). Most of the CD24$^+$ γδ T cells were *Sox13*$^+$, while the CD24$^-$ subset mainly consisted of *Il2rb*$^+$ cells (Figs 1C, 2C and EV3B). We identified four different clusters (1, 3, 20, and 23) in the CD24$^+$ *Sox13*$^+$ compartment. Cluster 1 expressed high levels of granzyme A (*Gzma*) representing a previously unknown sub-type (Figs 2C and EV3B). *Gzma* induces a pro-inflammatory cytokine response (Metkar *et al*, 2008), and *Gzma* produced by human Vγ9Vδ2 T cells inhibits growth of intracellular mycobacteria (Spencer *et al*, 2013). Therefore, this cluster may portray a novel cytotoxic sub-population of γδ T cells. Cluster 20 corresponds to another unknown sub-type best characterized by lower levels of *Sox13* and the co-expression of *Ccr9* and *S1pr1,* which are crucial for egress from the thymus and migration to peripheral sites (Figs 2C and EV3B) (Uehara *et al*, 2002; Matloubian *et al*, 2004). Except for this sub-type, the expression domains of *Ccr9* and *S1pr1* were more restricted to *Sox13*$^+$ and *Il2rb*$^+$ γδ T cells, respectively. It is thus likely that these cells may exit the thymus in a potentially naïve state and are polarized in the periphery. Clusters 3 and 23 express γδT17-associated genes such as *Sox13*, *Blk*, *Rorc*, *Il17a*, and *Il17f* (Figs 2C and EV3B). The CD24$^-$ compartment (clusters 6 and 10) mainly consisted of *Il2rb*$^+$ cells (Figs 2C and EV3B) representing IFN-γ-producing γδ T cells. Other genes exclusively expressed by clusters 6 and 10 included *Gzmb*, *Nt5e,* and *Klrb1a*. Cluster 10 exhibited strong expression of interferon-induced transmembrane protein family member *Ifitm1* and protein tyrosine kinase-binding protein *Tyrobp* (Figs 2C and EV3B). In summary, we characterized five fetal γδ sub-types—*Gzma*$^{hi}$, *Rorc*$^+$ *Il17a*$^+$ *Il17f*$^+$, *S1pr1*$^+$ *Ccr9*$^+$, *Il2rb*$^+$, and *Ifitm1*$^+$ γδ T cells (Fig 2B).

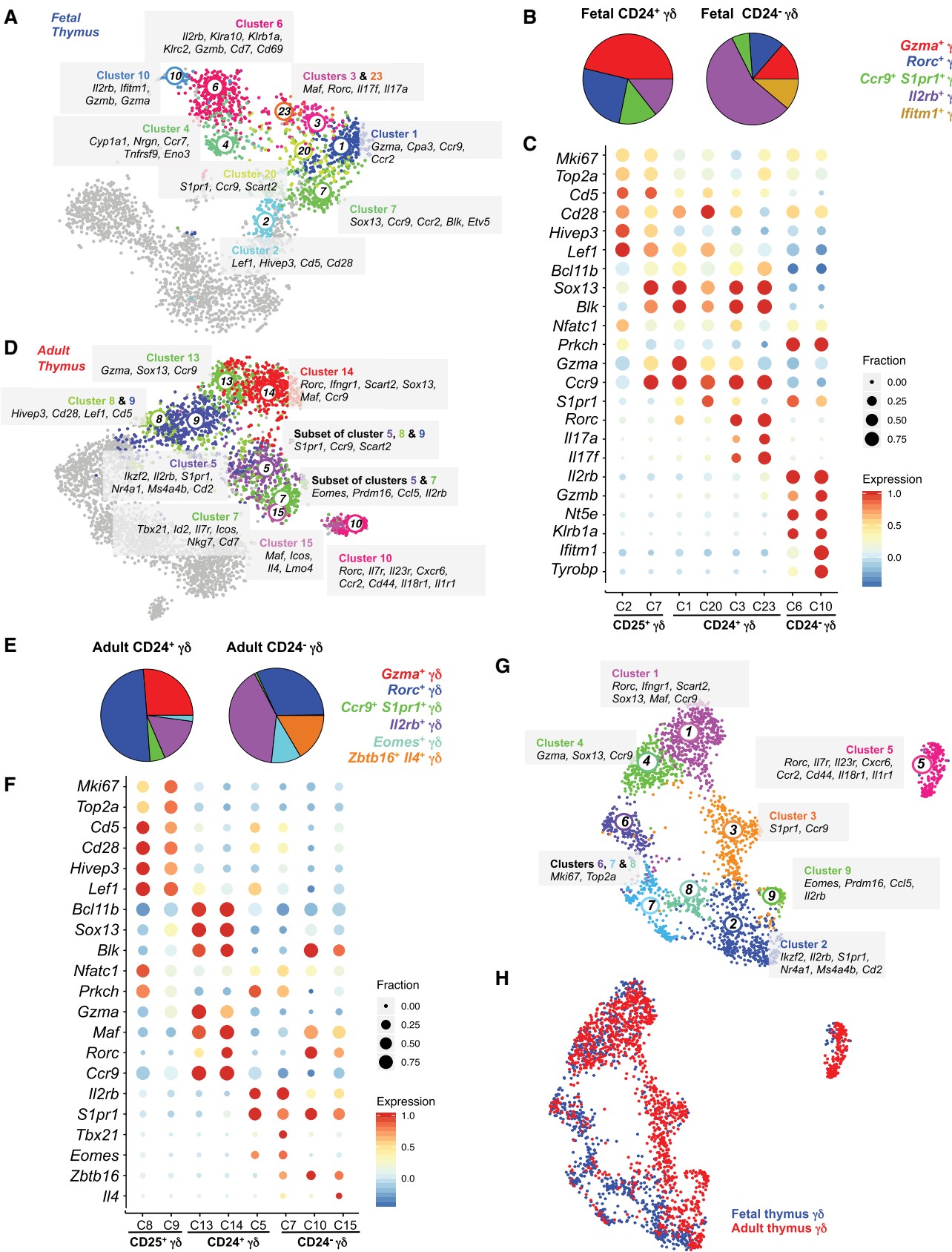

**Figure 2.**

Figure 2. Fetal and adult γδ T cells exhibit substantial transcriptional heterogeneity.

A t-SNE representation highlighting the fetal clusters enriched in immature and mature γδ T cells. Other cells are shown in gray. Selected marker genes characterizing these clusters are also depicted.

B Pie charts showing the contribution of the five identified sub-types to the immature (CD24+) and mature (CD24−) γδ T-cell compartment in the fetal thymus.

C Dot plot showing key marker genes differentially expressed among various γδ T cell sub-types. Color represents the z-score of the mean expression of the gene in the respective cluster and dot size represents the fraction of cells in the cluster expressing the gene. z-scores above 1 and below -1 are replaced by 1 and -1, respectively.

D t-SNE representation highlighting the adult clusters enriched in immature and mature γδ T cells. Other cells are shown in gray. We identified six different sub-types. Selected marker genes characterizing these clusters are also depicted.

E Pie charts showing the contribution of six identified sub-types to the immature (CD24+) and mature (CD24−) γδ T-cell compartment in the adult thymus.

F Dot plot showing key marker genes differentially expressed among various γδ T cell sub-types. Color represents the z-score of the mean expression of the gene in the respective cluster and dot size represents the fraction of cells in the cluster expressing the gene. z-scores above 1 and below −1 are replaced by 1 and −1, respectively.

G, H Uniform Manifold Approximation and Projection (UMAP) representation (G) showing 9 clusters identified in the integrated fetal and adult dataset. Few marker genes characterizing these clusters are also shown and (H) highlights the fetal and adult cells in blue and red color, respectively.

The adult immature CD24+ immature compartment comprises clusters 5, 13, and 14 (Figs 1G and 2D). Akin to fetal cluster 1, adult cluster 13 expresses high levels of *Gzma*, indicating that this subset is maintained in adults (Figs 2F and EV3C). Cluster 14, expressing *Sox13*, *Maf*, and *Rorc*, represents an immature γδT17 subset (Figs 2F and EV3C). Unlike fetal *Rorc+* cells, this cluster does not express *Il17a* and *Il17f*, supporting the hypothesis that fetal γδT17 cells are rapid "natural" IL-17 producers, while adult γδT17 cells are "inducible" IL-17 producers requiring antigen exposure in secondary lymphoid organs (Chien *et al*, 2013). Cluster 5 comprises CD122+ immature cells expressing *Il2rb* and represents the IFN-γ-producing lineage (Figs 2F and EV3C). Two transcriptionally distinct sub-types within cluster 5 were not resolved by clustering—*Eomes+* and *S1pr1+Ccr9+* cells (Figs 2F and EV3C). Of note, we found very few *Eomes+* cells in the fetal thymus. Similar to fetal cluster 20, we identified *Ccr9* and *S1pr1* double positive cells in adult cluster 5 (Figs 2F and EV3C). Few cells from clusters 8 and 9 were also *S1pr1+Ccr9+* (Fig 2D). Although these double positive cells seem to be transcriptionally similar to *Il2rb+* cells, they also expressed *Sox13* and *Scart2* associated with γδT17 differentiation, albeit at lower levels than in the fetal thymus. Overall, we identified five transcriptionally unique immature γδ subsets in the adult thymus—*Gzma*hi, *Rorc+*, *S1pr1+Ccr9+*, *Il2rb+*, and *Eomes+* subsets (Fig 2E).

Next, we characterized the transcriptional heterogeneity among the mature CD24− cells—a rare population in the adult thymus. Broadly, mature γδ T cells expressed one of four transcription factors—*Tbx21* (cluster 7), *Eomes* (a subset of cluster 7), *Rorc* (cluster 10), and *Zbtb16* (cluster 15 and a subset of cluster 10; Figs 2F and, EV3C and D). *Tbx21* is required for the differentiation of IFN-γ-producing cells (Barros-Martins *et al*, 2016) and, accordingly, was detected in cluster 7, which mainly comprises mature *Il2rb+* cells (Fig 2F). Cluster 10 comprised two subpopulations of *Rorc+* cells demarcated by the expression of two scavenger receptors—*Scart1* and *Scart2* (Fig EV3D). *Scart1+* cells expressed *Zbtb16* (encoding PLZF), required for the differentiation of Vγ6+ γδT17 cells developed in the fetus (Lu *et al*, 2015). Recently, *Scart1* has been shown to be expressed by tissue-resident Vγ6+ γδ T cells, strengthening the assumption that these cells are Vγ6+ γδ T cells (Tan *et al*, 2019). *Zbtb16* was also expressed by cluster 15 together with *Il4*, indicating that these cells are IFN-γ/IL-4-producing NKT-like cells (Figs 2F and EV3C) (O'Brien & Born, 2010). Taken together, scRNA-seq revealed substantial heterogeneity of mature γδ T cells in the adult thymus and identified *Tbx21+*, *Eomes+*, and *Zbtb16+Il4+* sub-types in the

*Il2rb+* compartment, and two sub-types in the *Rorc+* compartment, one of which was *Zbtb16+* (Fig 2E). Next, we utilized quadratic programming to identify the corresponding cell types in fetal and adult datasets. Fetal *Gzma*hi cluster 1, *Rorc+* clusters 3 and 23, and *Il2rb+* cluster 6 showed transcriptional similarities to adult *Gzma*hi cluster 3, immature *Rorc+* cluster 14, and *Il2rb+* cluster 7, respectively (Fig EV3E). We did not identify adult clusters corresponding to fetal cluster 10 (*Ifitm1+*) or fetal clusters equivalent to adult mature *Rorc+* cluster 10, indicating that these populations are developmental stage-specific.

To further investigate the correspondences among the developing γδ sub-types in the fetal and adult thymus, we integrated the fetal and adult γδ thymocyte datasets using a strategy described previously (Stuart *et al*, 2019) and inferred nine clusters in the integrated dataset (Fig 2G). Clusters 1 (immature γδT17), 2 (*Il2rb+*), and 4 (*Gzma*hi) were comprised of cells from both fetal and adult γδ thymocytes (Figs 2G and H, and EV3F). Clusters 6, 7, and 8 were characterized by the expression of cell cycle-related genes and mainly contained fetal γδ thymocytes indicating—akin to fetal DN cells—fetal γδ thymocytes are more proliferative (Figs 2G and H, and EV3F). As observed in the separate analysis, *Ccr9+ S1pr1+* (cluster 3) and mature γδT17 (cluster 5) sub-types were more abundant in the adult thymus (Figs 2G and H, and EV3F). *Eomes+* cells (cluster 9) were exclusively present in the adult thymus (Figs 2G and H, and EV3F). In conclusion, the integrated analysis of the fetal and adult γδ thymocytes reveals shifts in relative abundances of different sub-types at these two developmental time points.

### *Ccr9+ S1pr1+* γδ T cells represent a major subset of blood and lymph node γδ T cells and produce IFN-γ, TNF-α, and IL-2 upon stimulation

After development, γδ T cells exit the thymus and enter the periphery. Therefore, we characterized the transcriptional signature of circulating γδ T cells in the peripheral blood of the adult mouse. We sorted pan γδ T cells from the peripheral blood for scRNA-seq and classified circulating γδ T cells into seven clusters (Fig 3A). Cluster 6 expressed γδT17-associated genes such as *Blk*, *Maf*, and *Rorc* (Fig 3B). Circulating *Il2rb+* γδ T cells were more heterogeneous and segregated into clusters 1, 3, and 7 (Fig 3A and B). Cluster 3 exhibits exclusive expression of *Gzma* and *Gzmb*, while cluster 7 exclusively expresses *Ly6c2* and upregulates *S1pr1* (Fig 3B). Both clusters express *Ccl5*. Cluster 1 exhibits higher level of *Id2* and *Ccr2* (Fig 3B).

Importantly, cluster 8 represented by ~ 35% of circulating γδ T cells is characterized by the co-expression of *Ccr9* and *S1pr1*, similar to the novel naïve γδ subset in the embryonic and adult thymus (Fig 3A–D). Since *Ccr9*⁺ *S1pr1*⁺ γδ T cells did not exhibit either γδT17-associated or IFN-γ-producing γδ T-cell-associated gene expression signatures and rarely express *Cd44* (Fig 3C), we hypothesized that these cells are potentially naïve and CD44⁻. Therefore, we stratified blood γδ T cells into three different sub-types based on the expression of CD44 and CD122, i.e., CD44⁻, CD44⁺ CD122⁻, and CD44⁺ CD122⁺ and sorted them using FACS to perform scRNA-seq (Fig 3E). Combined analysis of pan γδ T cells and the sorted subsets revealed that the CD44⁻ gate mainly contained *Ccr9*⁺ *S1pr1*⁺ γδ T cells (Fig 3F). scRNA-seq analysis of *Ccr9*⁺ *S1pr1*⁺ γδ T cells (sorted as CD44⁻) after phorbol 12-myristate 13-acetate (PMA) and ionomycin stimulation revealed that these cells highly express several cytokines such as *Ifng*, *Tnf*, and *Il2* compared to unstimulated cells (Fig 3G–I). Our results suggest that *Ccr9*⁺ *S1pr1*⁺ γδ T cells are a subset of IFN-γ-producing γδ T cells but exit the thymus in an immature state and are polarized in the periphery in an adaptive-like fashion. Since the accumulation of *Ccr9*⁺ *S1pr1*⁺ γδ T cells in the circulation may reflect their incapability to migrate to the peripheral organs, we investigated the presence of this sub-population in the lymph nodes of adult mice. A combined analysis of blood and lymph node γδ T cells revealed that indeed ~ 55% of lymph node γδ T cells are *Ccr9*⁺ *S1pr1*⁺ (Fig 3J–L, Appendix Fig S1A and B).

In order to further compare the peripheral blood and lymph node γδ subsets with their thymic counterparts, we integrated scRNA-seq datasets of γδ T cells from the adult thymus, peripheral blood, and lymph nodes (Fig 3M and N). γδT17 and *Il2rb*⁺ γδ T cells from the blood and lymph nodes co-clustered with mature γδT17 (cluster 3) and *Il2rb*⁺ γδ thymocytes (clusters 4, 5, and 8), respectively (Fig 3M and N and Appendix Fig S1C). Cluster 2 mainly consisted of *Gzma*ʰⁱ and immature γδT17 thymocytes. These cells are absent in the blood and lymph nodes (Fig 3M, N and Appendix Fig S1C). Importantly, *Ccr9*⁺ *S1pr1*⁺ γδ T cells from the blood and lymph nodes also co-clustered with their thymic counterparts (cluster 1; Fig 3M, N, and Appendix Fig S1C). Overall, our analysis revealed common and unique γδ sub-types in the thymus, blood, and lymph nodes. We identified an unpolarized *Ccr9*⁺ *S1pr1*⁺ sub-population of γδ T cells in the thymus which lack the expression of known marker genes of γδT17 and IFN-γ-producing γδ T-cell lineages and produce several cytokines upon stimulation. In comparison with the adult thymus, this sub-population is expanded in the peripheral blood and lymph nodes, indicating that these cells are preferentially recruited from the thymus to these tissues.

### Characterizing γδ T cells expressing different variable chains

We next profiled γδ T cells expressing distinct TCR variable chains, which correspond to discrete tissue migration patterns of these cells. We profiled Vγ1⁺, Vγ4⁺, and Vγ5⁺ cells from the fetal and Vγ1⁺, Vγ4⁺, and Vγ1⁺ Vδ6.3⁺ cells from the adult thymus (Heilig & Tonegawa, 1986), isolated from the immature and mature compartments using scRNA-seq (Fig EV4A and B). Fetal Vγ4⁺ cells were sorted without using CD24 as most of them were CD24⁺. We classified our original fetal dataset sorted without variable chain information into three most abundant sub-types, i.e., *Gzma*⁺, *Rorc*⁺, and *Il2rb*⁺ cells

and assigned fetal Vγ1⁺, Vγ4⁺, and Vγ5⁺ cells to these subsets. Immature and mature Vγ1⁺ cells were mainly *Gzma*⁺ and *Rorc*⁺ with a minor increase in the fraction of *Il2rb*⁺ cells in the mature Vγ1⁺ compartment (Fig EV4A, C and I). Immature Vγ5⁺ cells contributed equally to *Gzma*⁺, *Rorc*⁺, and *Il2rb*⁺ subsets but were overwhelmingly *Il2rb*⁺ in the mature state (Fig EV4A, D and I). Vγ4⁺ cells were mainly *Gzma*⁺ and *Rorc*⁺ (Fig EV4A, E and I). In the adult thymus, we classified γδ T cells sorted without variable chain information into five sub-types, i.e., *Gzma*⁺, *Rorc*⁺, *Il2rb*⁺, *Eomes*⁺, and *Zbtb16*⁺ *Il4*⁺ cells, and assigned adult Vγ1⁺, Vγ4⁺, and Vγ1⁺ Vδ6.3⁺ to these sub-types. Approximately half of the immature Vγ1⁺ cells were *Il2rb*⁺, and few were *Zbtb16*⁺*Il4*⁺ (Fig EV4B, F and J). We also found a fraction of *Gzma*⁺ and *Rorc*⁺ cells in the immature Vγ1⁺ compartment but most mature Vγ1⁺ cells were *Il2rb*⁺, *Eomes*⁺, and *Zbtb16*⁺ *Il4*⁺ (Fig EV4B, F and J). Immature Vγ4⁺ cells were *Gzma*⁺ and *Rorc*⁺, and after maturation, they upregulated *Rorc* (Fig EV4B, G and J). Vγ1⁺ Vδ6.3⁺ cells are NKT-like cells associated with IFN-γ and IL-4 production (Kreslavsky *et al*, 2009; O'Brien & Born, 2010). Immature Vγ1⁺ Vδ6.3⁺ cells comprise *Il2rb*⁺ and *Zbtb16*⁺*Il4*⁺ cells, while their mature counterpart is mainly *Zbtb16*⁺ *Il4*⁺ (Fig EV4B, H and J). This analysis indicates that subsets expressing distinct TCR variable chains are more heterogeneous in the immature state and become more focused toward a particular sub-type upon maturation.

### Inferring differentiation trajectories and GRNs of fetal and adult γδ T-cell development

In order to understand the regulatory control of γδ T-cell differentiation, we inferred differentiation trajectories with StemID2 (Grün *et al*, 2016; Herman *et al*, 2018) for two major γδ sub-types—γδT17 (*Rorc*⁺) and IFN-γ-producing (*Il2rb*⁺) subsets. Self-organizing maps (SOMs) of pseudo-temporal gene expression profiles along the predicted fetal γδT17 and IFN-γ-producing differentiation trajectories yielded 17 and 20 gene modules, respectively (Fig EV5A and C). Similarly, gene expression profiles along adult γδT17 and IFN-γ-producing differentiation trajectories were grouped into 22 and 31 gene modules, respectively (Fig EV5G and H). These modules were activated at distinct time points during differentiation. We focused on the late modules activated during effector differentiation into γδT17 and IFN-γ-producing lineages and identified many genes including transcription factors with known and unknown functions and with distinct dynamics (Fig EV5B, D–F, I and J). GSEA revealed that chromatin modifiers are specifically upregulated during γδT17 differentiation, whereas the IFN-γ lineage is more proliferative in the fetal and adult thymus (Fig EV5K and L). In particular, pseudo-temporal modules activated during effector differentiation of fetal and adult cells into the γδT17 lineage contained many histone-modifying factors such as histone-lysine *N*-methyltransferases *Kmt2a* and *Kmt2c*, lysine-specific demethylases *Kdm5a* and *Kdm5b*, and the histone deacetylase *Hdac7* (Fig EV5M).

In order to investigate how these differentially expressed genes are integrated in fetal or adult GRNs, we applied the random forests-based GENIE3 algorithm (Huynh-Thu *et al*, 2010). We recovered distinct GRN modules specifically activated at particular stages of γδ T-cell development (Fig 4A–D, Appendix Fig S2A and B). The earliest network comprises genes associated with ETPs (*Bcl11a*, *Cd44*,

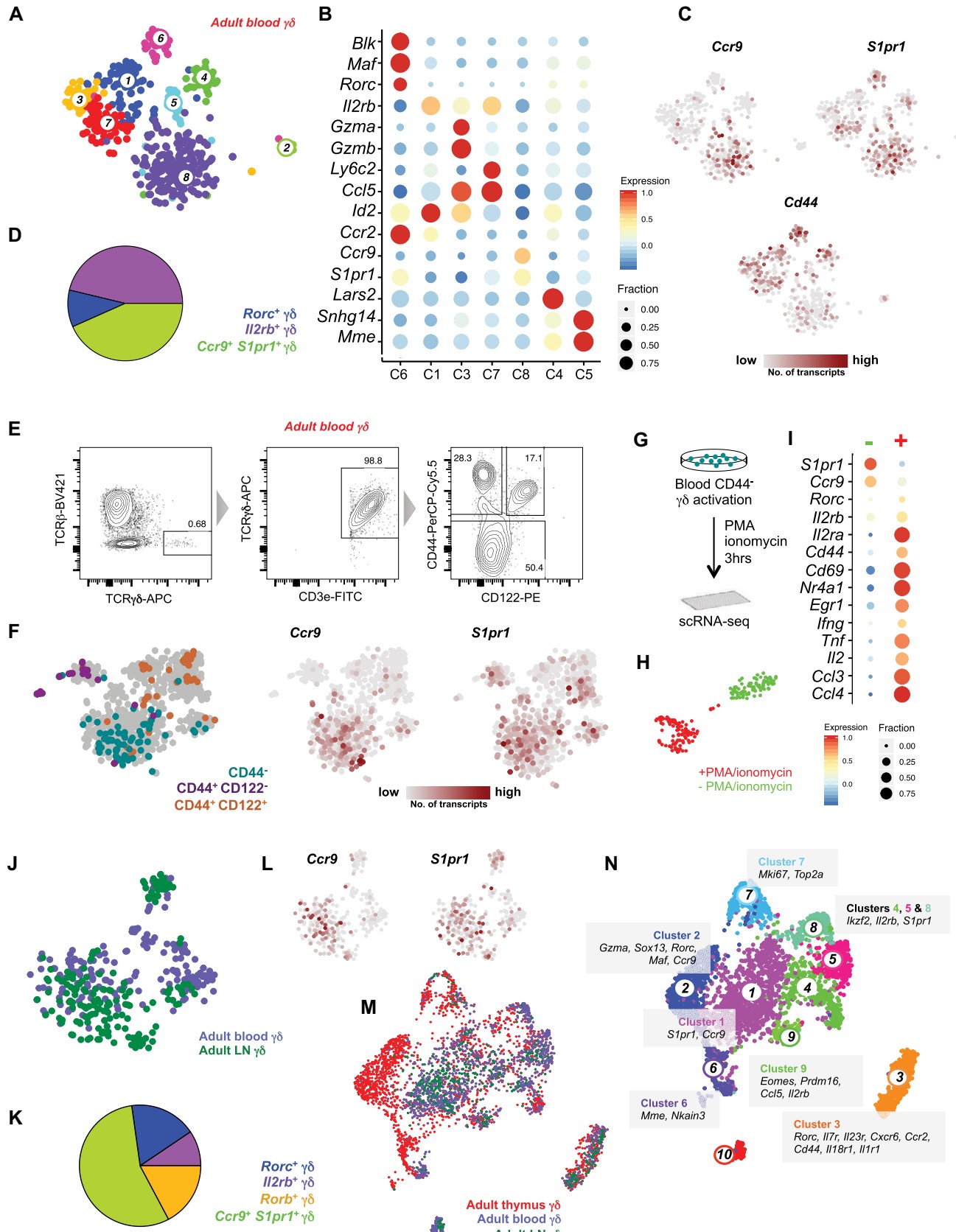

**Figure 3.**

**Figure 3.** *Ccr9⁺ S1pr1⁺* γδ T cells represent a major subset of blood and lymph node γδ T cells.

A t-SNE representation based on transcriptome similarities identified 8 clusters of circulating γδ T cells in peripheral blood (*n* = 2 independent experiments, twelve 6- to 7-week-old female mice).

B Dot plot showing key marker genes differentially expressed among circulating γδ T-cell sub-types.

C t-SNE representation showing the expression of *Ccr9, S1pr1*, and *Cd44*.

D Pie chart showing the fraction of three major γδ sub-types in the blood. Approximately 35% of circulating γδ T cells are *Ccr9⁺ S1pr1⁺*.

E FACS plots showing the gates used for sorting circulating γδ T cells based on the expression of CD44 and CD122.

F t-SNE representations of the combined analysis of CD44⁻, CD44⁺ CD122⁻, and CD44⁺ CD122⁺ γδ T cells (*n* = 1 independent experiment, six 6- to 7-week-old female mice) with pan γδ T cells from the peripheral blood (depicted in gray). Note that CD44⁻ γδ T cells are *Ccr9⁺ S1pr1⁺*.

G Schematic showing the workflow used for profiling the stimulated *Ccr9⁺ S1pr1⁺* γδ T cells.

H t-SNE representation showing the control and PMA/ionomycin-stimulated *Ccr9⁺ S1pr1⁺* γδ T cells (*n* = 2 independent experiments, twelve 6- to 7-week-old female mice).

I Dot plot showing key genes differentially expressed between the control and PMA/ionomycin-stimulated *Ccr9⁺ S1pr1⁺* γδ T cells.

J t-SNE representation of the combined analysis of blood (purple) and lymph node (green) γδ T cells (*n* = 1 independent experiment, lymph nodes from three 6- to 7-week-old female mice).

K Pie chart showing the fraction of three major γδ sub-types in lymph nodes. Approximately 55% of lymph node γδ T cells are *Ccr9⁺ S1pr1⁺*.

L t-SNE representation of the combined analysis of blood and lymph node gd T cells showing the expression of *Ccr9* and *S1pr1*.

M, N UMAP representation of the integrated dataset from the thymus, blood and lymph nodes (M) showing cells from different tissues in different colors and (N) showing 10 clusters identified in the integrated dataset. Few marker genes characterizing the clusters are also shown.

*Adgrg1, Kit* etc.) and T-cell commitment such as *Bcl11b, Cd3e, Cd3d,* and *Cd3e* and was thus termed core T-cell network. The subsequently activated network consists of genes involved in recombination, e.g., *Rag1, Rag2, Ptcra, Hdac4, Pld4,* and *Notch3*, and is, hence, addressed as recombination network. A third network was cell cycle-associated, comprising, e.g., *Mki67, Pcna,* and *Ran*. Finally, we observed a γδT17-associated network, involving *Sox13, Maf, Blk,* and *Rorc,* and an IFN-γ lineage network containing *Il2rb, S1pr1,* and killer cell lectin-like receptors such as *Klra10* and *Klrd1*. These five networks were recovered from the fetal and the adult thymus. Of note, the adult dataset gave rise to another network comprising *Cd44, Icos, Il7r, Il18r1, Ccr2,* and *Ccr6* activated at mature γδT17 stages, i.e., cluster 10 (Fig 4C and D and Appendix Fig S2B). Taken together, we successfully inferred GRNs and recovered differentially expressed gene networks characterizing distinct stages of γδ T-cell differentiation.

**Sequential activation of SOX13, c-MAF and RORγt is essential for γδT17 differentiation**

We next attempted to decipher the underlying molecular mechanisms of γδT17 differentiation. The pseudo-temporal gene expression changes along the predicted γδT17 differentiation trajectory in the fetal and adult thymus implicated an early role of *Sox13*, a subsequent activation of *Maf* (encoding c-MAF) and a late upregulation of *Rorc* during γδT17 differentiation (Fig 4G) and were supported by edges connecting these genes in the γδT17-specific network module (Fig 4E and F). Although the role of SOX13, c-MAF, and RORγt has been explored in γδT17 differentiation (Ivanov *et al*, 2006; Melichar *et al*, 2007; Gray *et al*, 2013; Malhotra *et al*, 2013; Barros-Martins *et al*, 2016; Zuberbuehler *et al*, 2019), their temporal dynamics and the mutual interplay in regulating γδT17 differentiation have not been studied in detail. To validate the predicted expression kinetics (i.e., *Sox13-Maf-Rorc*) and to investigate the function of these factors during γδT17 differentiation, we profiled the corresponding γδ T-cell subsets in *Sox13, Maf*, and *Rorc* knockout (KO) mice. We first sequenced immature and mature γδ T-cell subsets from the fetal and adult thymus of *Sox13* KO mice (Fig 5A and Appendix Fig S3A). In the fetal thymus, *Sox13* KO mice

lacked *Maf⁺ Rorc⁺ Il17a⁺ Il17f⁺* γδ T cells, indicating that *Sox13* acts upstream of *Maf* and *Rorc* to activate the γδT17-differentiation program (Fig 5B and C). Furthermore, we identified *Maf⁺ Gzma⁺* cells in the wild-type (WT) mice, which were absent in the *Sox13* KO mice (Fig 5B and C). In conclusion, *Maf⁺* cells were completely absent in the fetal *Sox13* KO thymus, suggesting that *Sox13* acts upstream of *Maf* to activate its expression. Accordingly, differential gene expression analysis between WT and KO cells revealed a sharp downregulation of *Maf* in *Sox13*-deleted cells, in addition to *Blk, Zbtb16,* and *Gzma* (Fig 5D). Genes related to TCR signaling such as *Cd28* and *Lck* were upregulated in the *Sox13*-deleted cells (Fig 5D), and GSEA between control and *Sox13*-deleted γδ T cells revealed that, irrespective of the γδ lineage, *Sox13*-deleted cells expressed higher levels of TCR signaling components and cell cycle-related genes (Fig 5E). These results demonstrate that *Sox13* activates *Maf* and directly or indirectly regulates TCR signaling during fetal γδ T-cell differentiation.

Next, we analyzed the effects of *Sox13* deletion on γδ T cells in the adult thymus, which were found to be less drastic but more diverse. In the immature compartment, *Sox13*-deleted cells clustered separately from WT cells (Fig 5F). Differential gene expression analysis revealed downregulation of genes required for γδT17 differentiation—*Maf, Blk,* and *Rorc*—in KO cells (Fig 5H), which were still expressed at lower levels in KO cells (Fig 5G). Adult *Sox13*-deficient immature γδ T cells expressed higher levels of *Cd28* and *Btla*, a receptor for the B7 homolog B7x (Fig 5H) (Watanabe *et al*, 2003). *Sox13*-deleted cells from fetal and adult thymus exhibited higher expression of *Itm2a*, which has been shown to be upregulated through TCR signaling and to follow similar kinetics as *Cd69* (Fig 5D and H) (Kirchner & Bevan, 1999). These observations suggest that *Sox13* may play a role in downregulating TCR signaling components at both developmental time points. Furthermore, *Sox13*-deleted cells exhibited reduced expression of *Gzma*, and in the mature compartment, we observed downregulation of γδT17-associated genes, such as *Maf, Blk*, and *Rorc,* and of markers of other sub-types, i.e., *Zbtb16, Nt5e, Ly6a,* and *Il4* (Fig 5H). Of note, *Zbtb16* was expressed in two mature subsets in the adult thymus— *Il4⁺* cells, closely resembling NKT-like Vγ1⁺ Vδ6.3⁺ γδ T cells, and *Scart1⁺ Rorc⁺* cells (Fig EV3C), most likely a fetal-derived Vγ6⁺

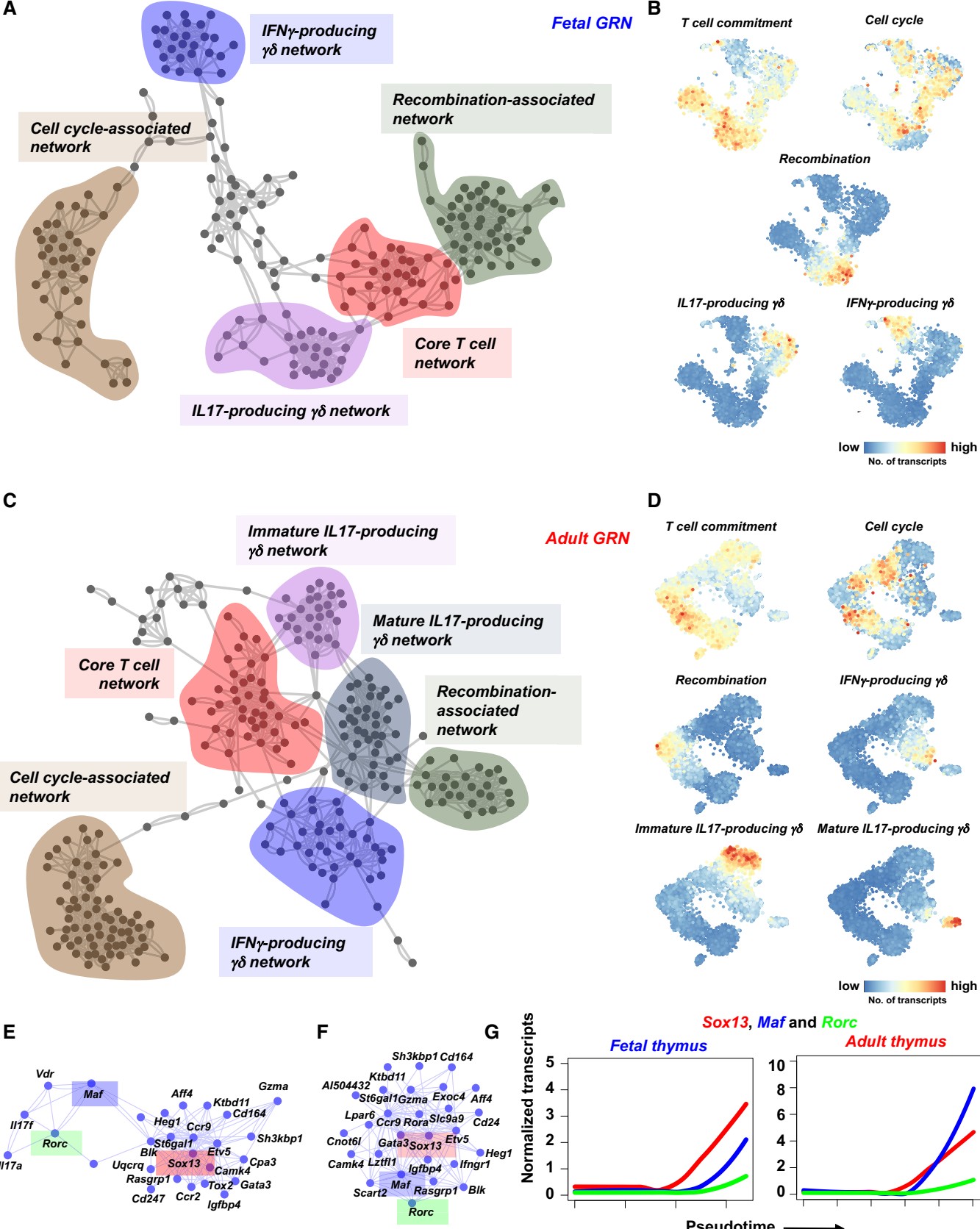

**Figure 4.**

**Figure 4.  Gene regulatory network (GRN) inference of fetal and adult γδ T-cell differentiation using scRNA-seq data.**

A   GRN as inferred from the fetal scRNA-seq data using the GENIE3 algorithm. The data of the top 1,000 interactions were used to construct the GRN. Recovered network modules are labeled and highlighted in different colors. Gene names in different networks are listed in Appendix Fig S2A.

B   t-SNE representation showing the aggregated expression of genes present in the different modules.

C   GRN as inferred from the adult scRNA-seq data using the GENIE3 algorithm. The data of the top 1,500 interactions were used to construct the GRN. Recovered network modules are labeled and highlighted in different colors. Gene names in different networks are listed in Appendix Fig S2B.

D   t-SNE representation showing the aggregated expression of genes present in the different modules.

E   Fetal γδT17 network. Note the presence of *Sox13*, *Maf* and *Rorc* (highlighted using rectangles with red, blue, and green colors, respectively). *Il17a* and *Il17f* are also part of this module.

F   Adult γδT17 network. Many genes were shared between the fetal and adult γδT17 network. *Sox13*, *Maf,* and *Rorc* are highlighted using rectangles with red, blue, and green colors, respectively.

G   Pseudo-temporal expression profiles of *Sox13*, *Maf* and *Rorc* along the fetal and adult γδT17 trajectory, respectively. Note the sequential order of expression along the pseudo-temporal order in both datasets.

γδT17 sub-type (Kreslavsky *et al*, 2009). Accordingly, we also observed reduced numbers of two *Zbtb16*[+] sub-types—*Il4*[+] and *Rorc*[+] cells (Fig 5F and G). In conclusion, we show that SOX13 acts upstream of c-MAF and PLZF and regulates the differentiation of γδT17 as well as NKT-like IFN-γ/IL-4-producing γδ T cells.

Our computational analysis predicted *Maf* as the link between *Sox13* and *Rorc* in specifying the γδT17 cell fate. In order to pinpoint the role of c-MAF in this regulatory hierarchy, we deleted *Maf* in lymphoid progenitors using *Il7ra^cre^*. We then performed scRNA-seq of immature and mature γδ T cells from the fetal and adult *Maf^fl/fl^; Il7ra^cre^* and *Maf^fl/fl^* thymi (Fig 5A and Appendix Fig S3B). We found a complete absence of *Rorc*[+] *Il17a*[+] *Il17f*[+] γδ T cells in the fetal KO thymi (Fig 5I and J). Accordingly, differential gene expression analysis within the immature fetal compartment revealed that KO cells downregulated *Gata3*, *Sox13*, *Blk*, *Icos*, *Rorc,* and *Il17f* (Appendix Fig S3C). In the adult thymus, immature cells from the KO mice clustered separately from WT controls (Fig 5K), with downregulation of *Sox13* and failure to express *Rorc* (Fig 5K and L, and Appendix Fig S3D). Differential gene expression analysis revealed that *Maf*-deleted cells from the immature compartment downregulated *Gata3*, *Blk,* and *Scart2* (Appendix Fig S3E). Importantly, *Maf*-deleted cells were found to upregulate the expression of TCR signaling strength-related genes such as *Nr4a1* and *Cd69* (Appendix Fig S3E) (Ashouri & Weiss, 2017). Accordingly, GSEA identified upregulation of AKT, MAP Kinase, and Toll-like receptor 4 (TLR) signaling pathways in the *Maf*-deleted immature compartment, suggesting a role of c-MAF in downregulating TCR signaling during γδT17 differentiation (Appendix Fig S3F). Furthermore, all mature *Rorc*[+] sub-types (*Zbtb16*[+] and *Ztbtb16*[−] populations) were missing in the KO thymi (Fig 5K and L). In conclusion, c-MAF acts downstream of *Sox13* and is required for the activation of the IL-17 program during γδT17 differentiation.

Since γδT17 cells have been shown to play an essential role in psoriasis-like skin inflammation and *Sox13*-mutant mice are protected from psoriasis-like dermatitis (Cai *et al*, 2011; Gray *et al,* 2013), we investigated whether the absence of γδT17 cells in the skin and skin draining lymph nodes of *Maf* KO mice (Appendix Fig S3G) makes them resistant to psoriasis-inducing stimuli. Indeed, we found that the application of imiquimod (IMQ) on the dorsal skin, an established psoriasis model (van der Fits *et al*, 2009; Flutter & Nestle, 2013), resulted in reduced skin inflammation in the *Maf* KO mice compared to WT mice (Fig 5M). Moreover, histological analysis of the dorsal skin of *Maf*-deficient mice did not show epidermal thickening pathology, a characteristic feature of psoriasis (Fig 5N

and O). Reverse transcription–polymerase chain reaction (RT–PCR) analysis of the IMQ-treated dorsal skin of *Maf* KO at day 7 revealed an absence of IL-17a mRNA compared to the treated WT (Appendix Fig S3H). Accordingly, the dorsal skin harbored significantly reduced numbers of RORγt[+] γδ T cells in the *Maf* KO mice after 6 days of consecutive IMQ application (Appendix Fig S3I). Taken together, *Maf*-deficient mice exhibit a drastic reduction in RORγt[+] γδT17 cells in the skin and skin draining lymph nodes and are protected from IMQ-induced psoriasis-like skin inflammation.

Finally, we analyzed the effect of *Rorc* deficiency on γδ T-cell differentiation, which we predicted to act downstream of *Sox13* and *Maf* based on our pseudo-temporal analysis (Fig 4G). We profiled immature and mature γδ T cells from the fetal and adult thymus of *Rorc* KO mice (Appendix Fig S4A and B). Like *Sox13* and *Maf*-deleted fetal thymi, *Rorc* deletion led to an absence of *Sox13*[+] *Maf*[+] *Il17f*[+] γδ T cells in the fetal thymus (Appendix Fig S4C–E). *Rorc*-deleted cells did not exhibit reduced *Sox13* or *Maf* expression (Appendix Fig S4D), suggesting that *Rorc* acts downstream of these two transcription factors and is the end-point of γδT17 differentiation in the fetal thymus. In the adult thymus, immature cells from *Rorc* KO mice clustered separately, did not activate the γδT17-differentiation program and lacked *Il17re* expression (Appendix Fig S47F and G). Differential gene expression analysis revealed that *Rorc*-deleted cells downregulate *Maf* and *Scart2* expression (Appendix Fig S4I). Akin to the fetal thymus, the expression of *Sox13* remains unaffected in adult *Rorc*-deleted immature γδ T cells (Appendix Fig S4G). The mature γδ T-cell compartment completely lacked γδT17 cells in the *Rorc* KO mice (Appendix Fig S4F and H), suggesting a block in γδT17 maturation. In summary, during γδT17 development in the fetal and adult thymus, SOX13 acts as an upstream regulator of γδT17 lineage specification and is essential for c-MAF-driven activation of RORγt, a transcription factor essential for the induction of the IL-17 program.

## Discussion

The γδ T-cell lineage has sparked a significant interest because of its unique developmental paradigm and functional properties, which can be exploited for immunotherapy. Two recent studies have also profiled γδ T cells from the murine thymus using scRNA-seq. However, the aims of these studies were not to understand intrathymic γδ T-cell differentiation. Ravens and colleagues explored the adaptation of Vγ6[+] γδ T cells to the skin and therefore

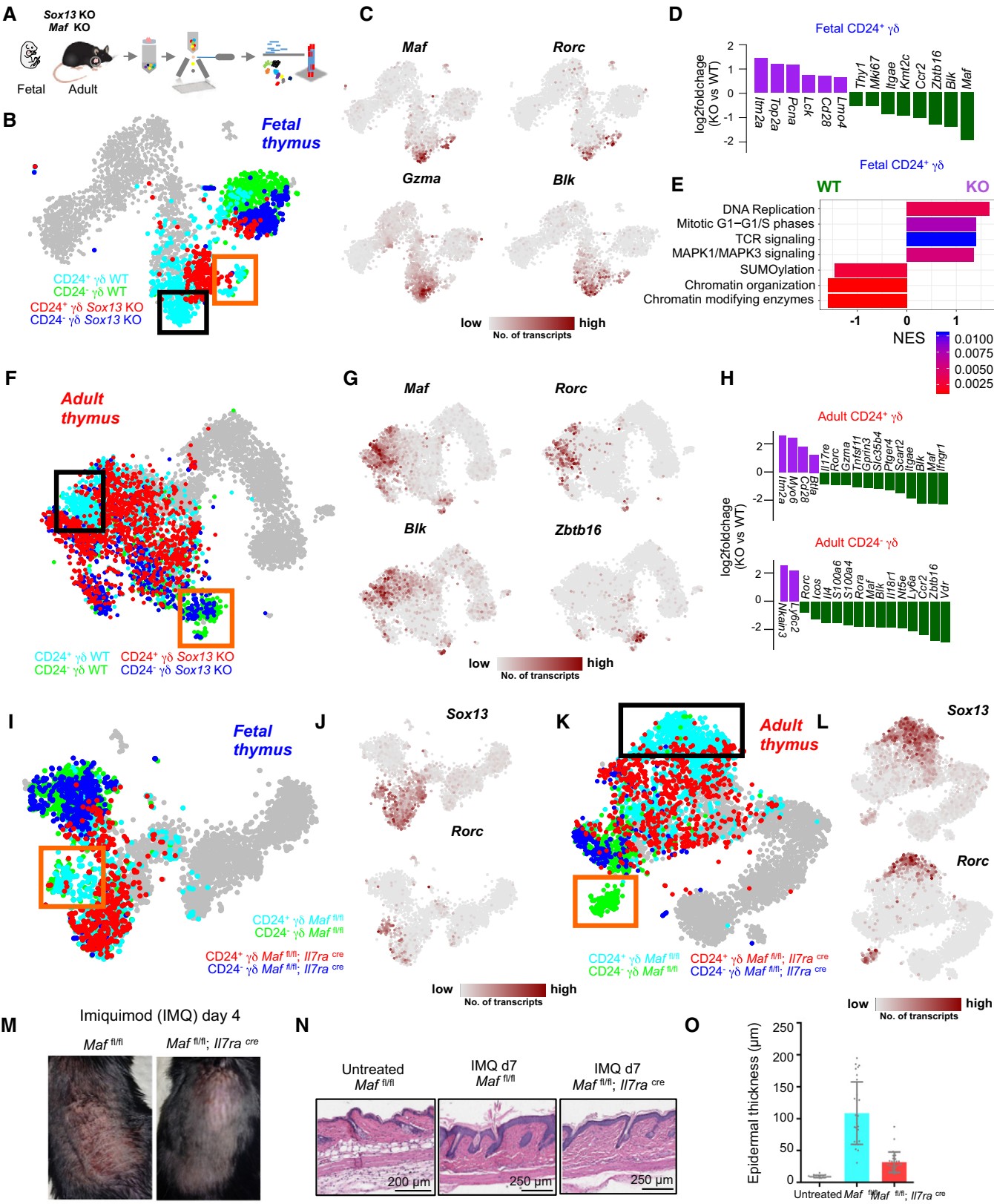

Figure 5.

**Figure 5.  Sequential activation of *Sox13* and *Maf* is required for the development of *Rorc*⁺ γδ T cells.**

A  Scheme showing the experimental design and scRNA-seq pipeline for the analysis of *Sox13* and *Maf* KO mice.

B  t-SNE representation of fetal cell types. Colors represent sorted cell types. Gray color represents DN and CD25⁺ γδ T cells from the fetal WT data shown in Fig 1C. γδT17 cells expressing *Maf*, *Rorc*, *Il17a*, and *Il17f* (orange box) as well as *Maf*⁺ (black box) γδ T cells were absent in the *Sox13* KO fetal thymus (n = 2 independent experiments, twelve embryos from two female mice per genotype).

C  t-SNE representation highlighting the expression of *Maf*, *Rorc*, *Gzma*, and *Blk*.

D  Bar plot depicting the differentially expressed genes in immature γδ T cells between the *Sox13* KO and WT fetal thymi (purple: upregulated genes, green: downregulated genes, adjusted *P* < 0.05).

E  Gene set enrichment analysis (GSEA) of differentially expressed genes between immature γδ T cells from the *Sox13* KO and WT fetal thymi. KO cells were more proliferating and upregulated TCR signaling-related genes.

F  t-SNE representation of adult cell types. Colors represent sorted cell types. Gray represents DN and CD25⁺ γδ T cells from the adult WT data shown in Fig 1F. Immature KO cells (cyan) clustered separately from WT cells (red). γδT17 cells expressing *Maf* and *Rorc* (black box) were missing in the KO. Note that few KO cells expressed *Maf* and *Rorc* but at lower levels. Mature *Maf*⁺/*Rorc*⁺ as well as *Zbtb16*⁺ (orange box) γδ T cells were reduced in the *Sox13* KO fetal thymi (n = 3 independent experiments from three male mice, each genotype).

G  t-SNE representation highlighting the expression of *Maf*, *Rorc*, *Blk*, and *Zbtb16*.

H  Bar plot depicting the differentially expressed genes in immature and mature γδ T cells between the *Sox13* KO and WT adult thymi. (purple: upregulated genes, green: downregulated genes, adjusted *P* < 0.05).

I  t-SNE representation showing the sorted cell types from WT and *Maf* KO fetal thymi (n = 2 independent experiments, eight embryos from two female mice per genotype). Gray color represents DN and CD25⁺ γδ T cells from the fetal WT data shown in Fig 1C. Note that cells expressing *Rorc*, *Il17a*, and *Il17f* are absent in the KO mice (orange box).

J  t-SNE representation highlighting the expression of *Sox13* and *Rorc*.

K  t-SNE representation showing the sorted cell types from WT and *Maf* KO adult thymi (n = 3 independent experiments from three female mice per genotype). Gray color represents cells from the adult WT data shown in Fig 1F. Note that immature KO cells clustered separately from WT cells (black box) and that the mature *Rorc*⁺ γδ T-cell compartment lacks KO cells (orange).

L  t-SNE representation highlighting the expression of *Sox13* and *Rorc*.

M  Representative pictures of the dorsal skin of the 8-week-old female *Maf* KO and WT mice after 4 days of consecutive IMQ application.

N  Hematoxylin and eosin staining of the dorsal skin of untreated (WT) and treated (*Maf* KO and WT) mice.

O  Graph showing the quantification of epidermal thickness of untreated (WT) and treated (*Maf* KO and WT) mice (n = 1 independent experiment, three adult mice each genotype). Bars represent the mean values and error bars indicate standard error of mean (SEM).

specifically profiled Vγ6⁺ γδ T cells from different tissues including thymus (Tan *et al*, 2019). In the second study, Maehr and colleagues investigated thymus organogenesis and profiled all fetal and newborn thymic cell types including blood cells without focusing on γδ T cells (Kernfeld *et al*, 2018). Their dataset contained only 769 non-conventional lymphocytes including innate lymphocytes-likes cells and γδ T cells. The latter were resolved in only three sub-types —*Gzma*ʰⁱ, γδT17, and proliferating γδT cells.

Here, we present the first focused and comprehensive map of γδ T-cell differentiation in the thymus. Since fetal and adult γδ T-cell population exhibit functional differences, we chose to infer high-resolution differentiation trajectories at these two stages of life (Prinz *et al*, 2013). Unexpectedly, our analysis revealed that gene regulatory programs of γδ T-cell differentiation and the emerging sub-types were surprisingly similar in fetal and adult thymi. In particular, we identified previously unknown sub-types including an unpolarized *Ccr9*⁺ *S1pr1*⁺ population, which is expanded in the peripheral blood and the lymph nodes of adult mice (Fig 6). Upon stimulation, they upregulate the expression of various cytokines such as *Tnf*, *Il2*, and *Ifng*. Therefore, although these cells leave the thymus in a functionally immature state, they are already primed to produce IFN-γ and are likely to acquire their effector phenotype in the periphery through an adaptive-like mechanism.

Current studies favor a TCR-dependent two-step model of γδ T-cell commitment and subsequent effector differentiation. TCR signal strength has been shown to play an essential role—weak signals promote αβ commitment and strong signals induce the γδ T-cell fate. The subsequent effector differentiation into IL-17-, IFN-γ-, and IL-4-producing γδ lineages also requires varying levels of TCR signals. Weak signals promote the γδT17 lineage, whereas progressively stronger signals promote IFN-γ- and IL-4-producing γδ lineages,

respectively (Fahl *et al*, 2014; Zarin *et al*, 2015). However, several studies have reported conflicting results owing to the different experimental systems used to decipher the role of TCR signals in driving the differentiation of various γδ T-cell effector sub-types (Wencker *et al*, 2014; Munoz-Ruiz *et al*, 2016; Sumaria *et al*, 2017). Therefore, whether TCR signals play an instructive or a permissive role in γδ effector differentiation is still debated. For example, a study by Kang and colleagues has identified a TCR-independent Sox13-ECFP⁺ c-KIT⁻ CD24⁺ DN1d sub-type which gives rise to the γδT17 lineage (Spidale *et al*, 2018). The study found that c-KIT⁺ DN1 cells (ETPs) did not generate CCR6⁺ γδT17 cells, while a fraction of γδT17 cells generated from DN1d progenitors was CCR6⁺. In this study, we focused our analysis on c-KIT⁺ DN1 cells, i.e., ETPs, and did not profile c-KITˡᵒ/⁻ DN1 thymocytes. However, in the adult thymus we found a considerable difference in the expression profile of mature and immature γδT17 cells as reflected by their clear separation in the t-SNE map (Fig 1F). Moreover, trajectory analysis did not link these two cell populations together (Fig 1I). Consequently, we did not include mature γδT17 cells while reconstructing the γδT17 differentiation trajectory in the adult thymus. Moreover, immature γδT17 cells were *Ccr6*⁻, while mature γδT17 cells were *Ccr6*⁺ (Fig EV3C). Thus, it is likely that mature γδT17 cells in the adult thymus are derived from Sox13-ECFP⁺ DN1d progenitors described by Kang and colleagues, while the immature γδT17 cells observed in this study are ETP-derived, and likely undergo maturation in the periphery without recirculating to the thymus. scRNA-seq profiling of DN1a-e thymocytes and the progeny they generate in differentiation assays such as fetal thymus organ culture (FTOC) may uncover the heterogeneity and the developmental potential of these progenitor compartments.

We focused our attention on the regulatory mechanisms governing γδT17 differentiation. We validated the role of three

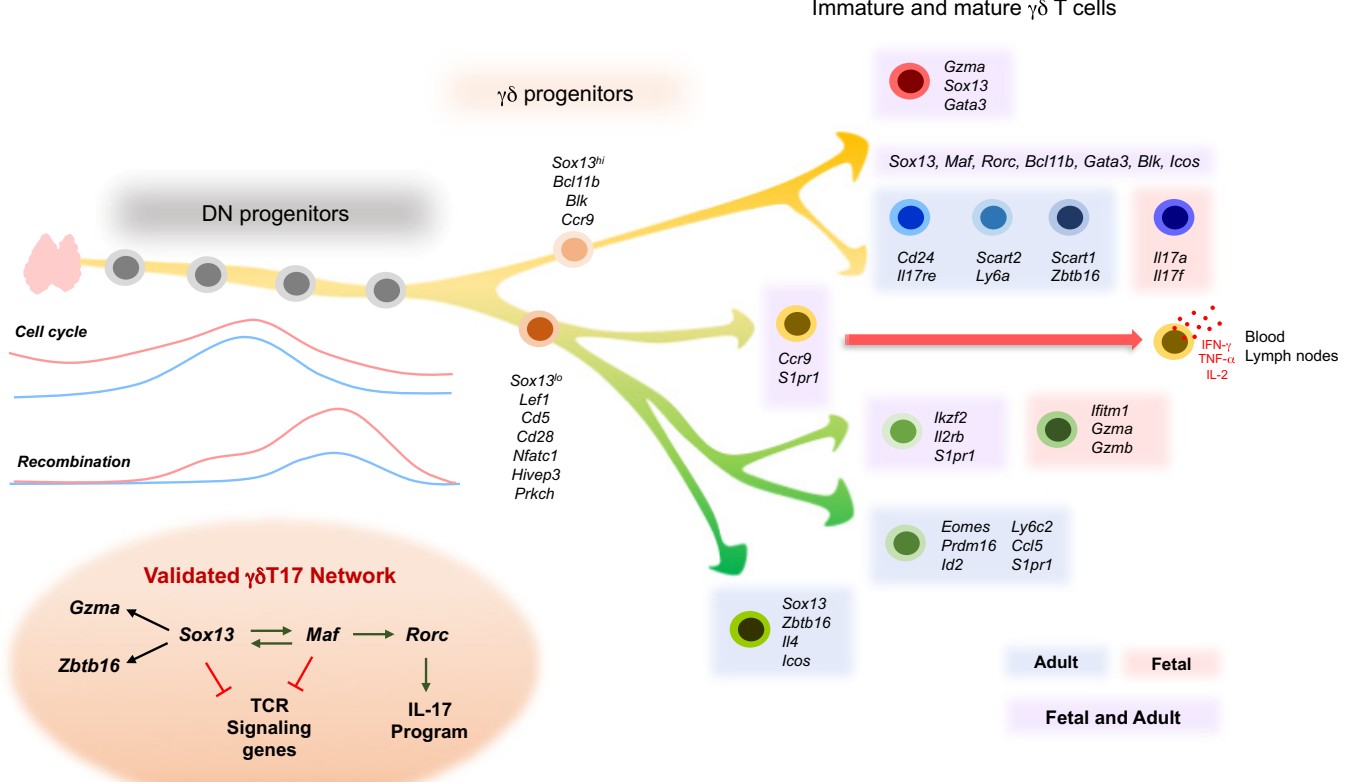

**Figure 6.  Transcriptional landscape of γδ T-cell development as revealed by scRNA-seq.**

scRNA-seq of early T-cell progenitors and γδ T cells during fetal and adult life reveals cell cycle and recombination-related differences in early thymopoiesis, continuous differentiation trajectories of γδ T-cell development, and identifies various γδ T-cell subsets at both developmental time points including an unpolarized *Ccr9*⁺ *S1pr1*⁺ population, which expands in the peripheral blood and lymph nodes and produces TNF-α, IFN-γ, and IL-2 upon stimulation. A combined analysis of *Sox13*, *Maf*, and *Rorc* KO mice reveals that sequential activation of these three transcription factors is essential for γδT17 commitment and differentiation.

transcription factors—*Sox13*, *Maf,* and *Rorc*—in sequentially driving the differentiation of γδ T-cell progenitors toward this lineage. However, the precise mechanistic details of the regulatory interactions among these three transcription factors remain to be investigated in future studies. Previously, it has been shown that Sox5 interacts with the DNA-binding domain of c-MAF via its HMG domain and cooperatively activates the promoter of RORγt in CD4⁺ T cells during T helper 17 cell differentiation (Tanaka *et al*, 2014). We therefore suspect a similar regulatory landscape where SOX13/SOX4 physically interacts with c-MAF to activate the RORγt promoter, leading to the activation of the γδT17-specific differentiation program. Chromatin accessibility and immunoprecipitation assays will be essential to dissect the intricacies of regulatory cascades driving the γδ T-cell differentiation program.

Interestingly, *Sox13* and *Maf*-deleted γδ T cells upregulated gene sets associated with enhanced TCR signaling strength. As mentioned above, several studies have shown that stronger TCR signals are required for γδ T-cell fate specification during αβ/γδ commitment. However, it remains unclear how a high TCR signal-experiencing γδ-committed progenitor downregulates these signals to differentiate into the γδT17 lineage. Our study hints toward a role of *Sox13* and *Maf* in this process. We noticed that *Sox13* and *Maf*-deleted cells

upregulate gene sets associated with TCR signaling components, indicating that these two transcription factors may play a role in downregulating the TCR signals in γδ-committed progenitors to drive differentiation of the γδT17 lineage. A recent study by Ciofani and colleagues has investigated the relationship between c-MAF and TCR signaling in detail by transducing *Rag1*-deficient DN thymocytes with transgenic γδTCR of varying signaling strength and observed an inverse relationship between them (Zuberbuehler *et al*, 2019). However, the effect of *Maf* deletion on TCR signaling was not specifically investigated. Consistent with our analysis, their study revealed an upregulation of PI3K, protein kinase A, and TCR signaling in *Maf*-deleted CD25⁻ CD27⁻ fetal γδT17 cells. Yet, it remains to be resolved whether *Sox13* and *Maf* directly or indirectly regulate TCR signaling.

Collectively, our results substantially expand the knowledge about the development of γδ T cells (Fig 6), an understudied branch of T cells at the interface of the adaptive and the innate immune system, which has lately received increasing attention as crucial player in autoimmune diseases and cancer. More broadly, our study provides the first detailed comparative analysis of cell states, gene networks, and differentiation trajectories during fetal and adult γδ T-cell development at single-cell resolution. Similar comparative studies to elucidate the differentiation of other immune cell types

across tissues or distinct stages of life will be essential to understand the differences in ontogeny and function over life and across tissues.

# Materials and Methods

### Mice

C57BL/6J mice were purchased from Charles River or obtained from in-house breeding. Mice were kept in the animal facility of the Max Planck Institute of Immunobiology and Epigenetics in specific-pathogen-free (SPF) conditions. All animal experiments were performed in accordance with the relevant guidelines and regulations, approved by the review committee of the Max Planck Institute of Immunobiology and Epigenetics and the Regierungspräsidium Freiburg, Germany. Generation and genotyping of *Sox13*-deficient (*Sox13* KO) mice have been described previously (Baroti *et al*, 2016). *Sox13* KO mice were on a C57BL/6J background. Experiments were approved by the responsible local committees and government bodies (University, Veterinäramt Stadt Erlangen & Regierung von Unterfranken). *Maf* KO mice were bred and maintained in the animal facility of the Skirball Institute (New York University School of Medicine) in SPF conditions. C57Bl/6 mice were obtained from Jackson Laboratories or Taconic Farm. *Maf*$^{fl/fl}$ and *Il7ra*$^{Cre}$ mice were kindly provided by C. Birchmeier and H. R. Rodewald (Schlenner *et al*, 2010; Wende *et al*, 2012). *Maf* conditional KO mice were generated by crossing *Maf*$^{fl/fl}$ to *Il7ra*$^{Cre}$ animals. All animal procedures were performed in accordance with protocols approved by the Institutional Animal Care and Usage Committee of New York University School of Medicine. Generation of *Rorc* KO (*Rorc*$^{tm2Litt}$) mice has been described previously (Eberl *et al*, 2004). *Rorc*$^{tm2Litt}$ mice were bred and maintained in the animal facility of Institute of Medical Microbiology and Hygiene at University Medical Center Freiburg and experiments were approved and are in accordance with the local animal care committees (Regierungspräsidium Freiburg).

### Thymocyte isolation

All animals were sacrificed using carbon dioxide and cervical dislocation. To isolate thymocytes, thymus was dissected and placed on a 40-μm cell strainer (Falcon, Corning) kept on a 50-ml tube (Falcon, Corning). Each adult thymus was mashed on the cell strainer using the back of the 1-ml syringe plunger. Fetal thymi from each pregnant mouse were pooled together and mashed on the cell strainer using the front of the 1-ml syringe plunger. Ten milliliter phosphate-buffered saline (PBS) was continuously added while mashing to collect the single-cell suspension of thymocytes in the 50-ml tube. Collected thymocytes were centrifuged at 400 *g* for 5 min at 4°C. The pellet was dissolved in 10 ml PBS and passed through the 30-μm nylon filter (CellTrics, Sysmex) kept on a 15-ml tube (Falcon, Corning). Cells were again centrifuged at 400 *g* for 5 min at 4°C. Afterward the pellet was dissolved in 200 μl of PBS.

### Lymph node γδ T-cell isolation

Isolation of the cells from the lymph nodes was performed as described above. Inguinal lymph nodes were used for the isolation. Lymph nodes from three mice were pooled together prior to isolation. The pellet was dissolved in 200 μl of PBS.

### Circulating γδ T-cell isolation

After sacrificing the animals using carbon dioxide and cervical dislocation, blood was quickly collected using cardiac puncture and placed in a 15-ml falcon tube containing 10 ml PBS. Blood from 3 to 6 mice was pooled together. Red blood cell lysis was performed using red blood cell lysis buffer (RBC lysis buffer, 10×, BioLegend) according to manufacturer's protocol. At the last step, the pellet was dissolved in 200 μl of PBS.

### Magnetic enrichment of adult double negative thymocytes

After thymocyte isolation as describe above, the pellet was dissolved in 3 ml of PBS. Three ml of the following biotin-labeled antibody cocktail solution (1:50 dilution each, clones are mentioned in the brackets) was prepared: CD8a (53–6.7), CD4 (GK1.5), NK-1.1 (PK136), TER-119 (TER-119), Ly-6G/Ly-6C (RB6-8C5), and CD11c (N418). All antibodies were purchased from BioLegend. The antibody solution was incubated with the thymocytes for 15 min on ice. The cells were centrifuged at 400 *g* for 5 min at 4°C and washed with 5 ml of PBS. The resulting pellet was resuspended in 600 μl of PBS, and 100 μl streptavidin-conjugated beads (MojoSort Streptavidin Nanobeads, BioLegend) were added to the solution and incubated for 15 min on ice. Afterward, the tube was placed on the magnet for 5 min to let the beads firmly attached to the wall of the tube in contact with the magnet. The remaining liquid containing the cells without the beads was transferred into a 1.5-ml tube (Eppendorf) and centrifuged at 400 *g* for 5 min at 4°C. The pellet was dissolved in 50 μl of PBS.

### Cryopreservation and thawing of thymocytes

Thymocytes from *Maf* KO and *Sox13* KO mice as well as the corresponding littermate controls were cryopreserved prior to shipment on dry ice to the Max Planck Institute of Immunobiology and Epigenetics, Freiburg. Thymocytes were isolated as described before and resuspended in 4 ml of freezing medium containing 90% fetal bovine serum (FBS) and 10% dimethyl sulfoxide (DMSO) and kept at −80°C overnight before shipment. After receiving the frozen thymocytes, they were kept at −80°C for few days. Prior to antibody staining, frozen cells (in 90% FBS and 10% DMSO) were de-frozen in the water bath at 37°C until almost thawed. The cells were transferred to a 15-ml tube (Falcon, Corning), and 14 ml of pre-warmed RPMI medium 1640 (Gibco) was added drop-wise while gently swirling and inverting the tube. The cells were then centrifuged at 400 *g* for 5 min at 4°C. The supernatant was discarded, and cells were washed again with 10 ml RPMI medium 1640. After centrifugation, the cell pellet was resuspended in 200 μl of RPMI medium 1640.

### Antibody staining, flow cytometry, and single-cell sorting

A 200 μl of antibody staining solution was prepared in PBS (for freshly isolated thymocytes and cells from the peripheral blood and lymph nodes) or RPMI medium 1640 (for frozen lymphocytes) and added to the 200 μl of resuspended pellet. For enriched DN

thymocytes, 50 μl of antibody solution was added to the pellet, which was dissolved in 50 μl of PBS. Afterward, thymocytes in the antibody solution were incubated for 20 min on ice. Cells were then washed twice with 1 ml of PBS or RPMI medium 1640 and resuspended in 3 ml after the last wash. Just prior to single-cell sorting using a flow cytometer, 5 μl of 20 μg ml$^{-1}$ 4′,6-diamidino-2-phenylindole (DAPI, Sigma) solution was added to the tube to stain for dead cells. Flow cytometry data were analyzed using the FlowJo program. The following antibodies were used (clones and used dilutions are mentioned in the brackets): CD117-BV510 (ACK2, 1:500), CD44-PerCP/Cy5.5 (IM7, 1:500), CD25-BV421 (PC61, 1:500), CD122-PE (TM-β1, 1:500), CD8a-BV421 (53-6.7, 1:1,000), CD8a-FITC (53-6.7, 1:500) CD4-PE (RM4-5, 1:1,000), CD4-APC/Cy7 (RM4-5, 1:500), CD4-APC (RM4-5, 1:500), TCRγδ-APC (GL3, 1:500), CD24-PE (M1/69, 1:1,000), CD24-PerCP/Cy5.5 (M1/69, 1:500), Vγ1.1 (2.11, 1:500), Vγ2-FITC (UC3-10A6, 1:500), Vγ3-PE (536, 1:500), and Vδ6.3/2-PE (8F4H7B7, 1:500). All the antibodies were purchased from BioLegend except CD8a-FITC, CD44-PerCP/Cy5.5, and Vδ6.3/2-PE (BD Pharmingen). Single cells were sorted in 384-well plates (Bio-Rad Laboratories) containing lysis buffer and mineral oil (see the next section) using BD FACSAria FUSION. The sorter was run on single-cell sort mode. Using pulse geometry gates (FSC-W × FSC-H and SSC-W × SSC-H), doublets were excluded. After the completion of sorting, the plates were centrifuged for 10 min at 2,200 *g* at 4°C, snap-frozen in liquid nitrogen, and stored at −80°C until processed.

### Stimulation and single-cell sorting of CD44⁻ circulating γδ T cells

6000 CD44⁻ circulating γδ T cells were sorted in 96-well plate in complete RPMI medium containing 10% FCS and stimulated for 3 h with Cell Activation Cocktail (with Brefeldin A; BioLegend). Non-stimulated 6000 CD44⁻ circulating γδ T cells cultured for 3 h in complete RPMI medium with 10% FCS served as control. Afterward, single cells were sorted in 384-well plates without antibody staining and processed for scRNA-seq as described below.

### Single-cell RNA amplification and library preparation

Single-cell RNA sequencing was performed using the mCEL-Seq2 protocol, an automated and miniaturized version of CEL-Seq2 on a mosquito nanoliter-scale liquid-handling robot (TTP LabTech) (Hashimshony *et al*, 2016; Herman *et al*, 2018). Twelve libraries with 96 cells each were sequenced per lane on Illumina HiSeq 2500 or 3000 sequencing system (pair-end multiplexing run) at a depth of ~ 130,000–200,000 reads per cell.

### Quantification of transcript abundance

Paired end reads were aligned to the transcriptome using bwa (version 0.6.2-r126) with default parameters (Li & Durbin, 2010). The transcriptome contained all gene models based on the mouse ENCODE VM9 release downloaded from the UCSC genome browser comprising 57,207 isoforms, with 57,114 isoforms mapping to fully annotated chromosomes (1–19, X, Y, M). All isoforms of the same gene were merged to a single gene locus. Furthermore, gene loci overlapping by > 75% were merged to larger gene groups. This procedure resulted in 34,111 gene groups. The right mate of each

read pair was mapped to the ensemble of all gene loci and to the set of 92 ERCC spike-ins in sense direction (Baker *et al*, 2005). Reads mapping to multiple loci were discarded. The left read contains the barcode information: The first six bases corresponded to the unique molecular identifier (UMI) followed by six bases representing the cell specific barcode. The remainder of the left read contains a polyT stretch. For each cell barcode, the number of UMIs per transcript was counted and aggregated across all transcripts derived from the same gene locus. Based on binomial statistics, the number of observed UMIs was converted into transcript counts (Grün *et al*, 2014).

### Clustering and visualization of thymocyte datasets

Clustering analysis and visualization were performed using the RaceID3 algorithm (Herman *et al*, 2018). The number of quantified genes ranged from 25,718 to 26,970 in different datasets. Cells with a total number of transcripts < 2,500 were discarded, and count data of the remaining cells were normalized by downscaling except fetal *Maf* KO and adult *Sox13* KO datasets where this cut-off was reduced to 1,500. Cells expressing > 2% of *Kcnq1ot1*, a potential marker for low-quality cells (Grün *et al*, 2016), were not considered for analysis. Additionally, transcripts correlating to *Kcnq1ot1* with a Pearson correlation coefficient > 0.65 were removed. The default parameters were used for RaceID3 analysis except probthr which was set to 10$^{-4}$. For the fetal data, ribosomal genes (small and large subunits) as well as predicted genes with *Gm*-identifier were excluded from the analysis. CGenes was initialized with the following set of genes to remove cell cycle-associated and batch-associated variability for the fetal data: *Malat1*, *Xist*, *Pcna*, *Mki67*, *Mir703*, *Ptma*, *Actb*, *Hsp90aa1*, *Hsp90ab1*, *Ppia* and *H19* and for the adult data: *Pcna*, *Mki67*, *Mir703*, *Gm44044*, *Gm22757*, *Gm4775*, *Gm17541*, *Gm8225*, *Gm8730*, *Ptma*, *Actb*, *Hsp90aa1*, *Hsp90ab1*, and *Ppia*. The analysis of γδ T cells expressing different variable chains was also performed with the same parameters as described above except for the adult dataset where cells with a total number of transcripts < 2,000 were discarded. Dimensionality reduction using t-SNE was performed using different values of the perplexity parameter. Overall, the structure of the data remained fairly stable across different values and the default value (set to 30) of the RaceID3 algorithm was used to represent both the datasets (Appendix Fig S5).

### Clustering and visualization of blood, lymph node, and stimulated datasets

For the circulating γδ T-cell analysis, cells with a total number of transcripts < 800 were discarded and count data of the remaining cells were normalized by downscaling. Cells expressing > 2% of *Kcnq1ot1*, a potential marker for low-quality cells (Grün *et al*, 2016), were not considered for analysis. Additionally, transcripts correlating to *Kcnq1ot1* with a Pearson correlation coefficient > 0.65 were removed. The default parameters were used for RaceID3 analysis except probthr which was set to 10$^{-4}$. Ribosomal genes (small and large subunits) as well as predicted genes with *Gm*-identifier were excluded from the analysis. The FGenes parameter was initialized with *Malat1* and *Xist*, and CGenes was initialized with the following set of genes: *Pcna*, *Mki67*, *Mir703*, *Ptma*, *Actb*, *Hsp90aa1*, *Hsp90ab1*, and *Ppia*. The combined data analysis of blood and lymph node γδ T cells was performed with the similar parameters

except that cells with a total number of transcripts < 500 were discarded. CGenes was initialized with the following set of genes: *Pcna*, *Mki67*, *Mir703*, *Ptma*, *Actb*, *Hsp90aa1*, *Hsp90ab1*, *Ppia, Ebf1, Slx1b,* and *Comt.* The PMA/ionomycin-stimulated dataset was analyzed with the similar parameters as the circulating γδ T-cell dataset but cells with a total number of transcripts < 500 were discarded. Default perplexity value (30) was used for the dimensionality reduction of the datasets using t-SNE.

### Combined analysis of γδ T cells from fresh and frozen thymocytes

In order to assess the effect of freezing and thawing on specific γδ subsets, we performed a combined analysis of fresh γδ T cells isolated from the mice which were kept in the animal facility of the Max Planck Institute of Immunobiology and Epigenetics with frozen γδ T cells from *Maf* and *Sox13* WT littermate controls at fetal and adult stages. For the combined fetal γδ T-cell analysis, cells with a total number of transcripts < 1,500 were discarded, and for the adult data, this threshold was set to 1,000. For both datasets, raw counts of the remaining cells were normalized using downscaling. Cells expressing > 2% of *Kcnq1ot1*, a potential marker for low-quality cells (Grün *et al*, 2016), were not considered for both analyses. Additionally, transcripts correlating to *Kcnq1ot1* with a Pearson correlation coefficient > 0.65 were removed. The default parameters were used for RaceID3 analysis except for probthr, which was set to $10^{-4}$. Ribosomal genes (small and large subunits) as well as predicted genes with *Gm*-identifiers were excluded from both analyses. The FGenes parameter was initialized with *Malat1*, *Xist*, *Ptma*, *Actb*, *Junb*, *Hsp90aa1*, *Hsp90ab1*, and *Ppia*. For the fetal data analysis, *Lrrc58* was also excluded from the clustering. We did not see any apparent effect of freezing on the transcriptome and recovered all γδ subsets from fresh as well as frozen material (Appendix Fig S6).

### Differential gene expression analysis

Differential gene expression analysis was performed using the diffexpnb function of RaceID3 algorithm. Differentially expressed genes between two subgroups of cells were identified similar to a previously published method (Anders & Huber, 2010). First, negative binomial distributions reflecting the gene expression variability within each subgroup were inferred based on the background model for the expected transcript count variability computed by RaceID3. Using these distributions, a *P*-value for the observed difference in transcript counts between the two subgroups was calculated and multiple testing corrected by the Benjamini–Hochberg method.

### Quadratic programming to quantify cell similarities between fetal and adult data

Using the quadratic programming approach, we calculated weights for all cluster medoids in one dataset for each cell in the other dataset using the solve.QP function of the quadprog R package. For example, to calculate the weights for all cluster medoids of the fetal data for each cell in the adult data, the adult normalized transcript count matrix containing genes as rows and single cells as columns and the fetal normalized transcript count matrix containing genes as rows and cluster medoids as columns were provided as inputs to

the Dmat and dvec arguments of solve.QP function, respectively. Cluster medoids were calculated by the compmedoids function of the RaceID3 algorithm. The intersect of feature genes expressed in both fetal and adult datasets was used for calculating the weights using quadratic programming.

### Lineage inference and pseudo-temporal ordering

To derive the differentiation trajectories of γδ T-cell differentiation, the StemID2 algorithm was used (Grün *et al*, 2016; Herman *et al*, 2018). StemID2 was run with the following parameters: cthr = 15, nmode = T, pthr = 0.01, pethr = 0.05, scthr = 0.6. Based on the adult γδT17 differentiation trajectory predicted by StemID2, cells of clusters 4, 2, 3, 6, 12, 11, 8, 9, 13, and 14 were used to compute the SOMs of pseudo-temporal expression profiles of the γδT17 lineage. Similarly, clusters 4, 2, 3, 6, 12, 11, 8, 9, 5, and 7 were used to compute the SOMs of pseudo-temporal expression profiles of the adult IFN-γ-producing lineage. Similarly, the following clusters were used to compute SOMs for the fetal γδT17 differentiation trajectory: 15, 14, 8, 13, 11, 5, 7, 1, 3, and 23; and for the IFN-γ differentiation trajectory: 15, 14, 8, 13, 11, 5, 2, 4, and 6.The fetal early DN differentiation trajectory was derived using the following clusters: 15, 14, 8, 13, 11, and 5; and for the adult dataset, the following clusters were used: 4, 2, 3, 6, 12, and 11.

### Gene regulatory network inference

Gene regulatory network inference was performed using the random forests-based ensemble method of the GENIE3 algorithm (Huynh-Thu *et al*, 2010). For the fetal GRN inference, only genes expressed at > 4 transcripts in at least two of the cells were included in the analysis. For the adult GRN inference, only genes expressed at > 5 transcripts in at least one of the cells were included in the analysis. R/C implementation of GENIE3 was used for network inference. The top 1,000 (fetal data) and 1,500 (adult data) regulatory links with highest weights were shortlisted for visualization of the network using Cytoscape with default settings (Shannon *et al*, 2003).

### Gene set enrichment analysis

Gene set enrichment analysis was performed using gsePathway function of ReactomePA, an R/Bioconductor package (Yu & He, 2016). The fold change for each gene was calculated between the cells to be compared using the diffexpnb function of RaceID3 and was given as an argument to gsePathway function to calculate enriched gene sets in KO cells using the following parameters: nPerm = 1,000, minGSSize = 120, pvalueCutoff = 0.05, pAdjustMethod="BH", organism = "mouse".

### Integration of scRNA-seq data

Integration of scRNA-seq datasets was performed using Seurat version 3 with default settings (Stuart *et al*, 2019). Integration analysis was performed on only those cells which were present in the separate analyses performed by the RaceID3 algorithm. Also, ribosomal genes (small and large subunits) as well as predicted genes with *Gm*-identifier were excluded from the data prior to integration.

## Skin inflammation models

The dorsal skin of 8-week-old mice in the telogen (resting) phase of the hair cycle was shaved with clippers and then subjected to topical application or treatment of the skin as below. IMQ: Mice were treated with either ~ 1 mg cm$^{-2}$ skin of 5% IMQ cream (Perrigo) or control Vanicream (Pharmaceutical Specialties Inc.) for six consecutive days as previously described (van der Fits *et al*, 2009).

## Cell isolation, tissue processing, and flow cytometry

Keratinocyte isolation was adapted from a previously described protocol (Nowak & Fuchs, 2009). In brief, dorsal skin was shaved and digested using either 0.25% trypsin/EDTA (Gibco) or collagenase (Sigma) to obtain a single-cell suspension. Immune cells from 1 cm$^2$ pieces of skin were isolated after digestion with liberase (Roche) based on an adapted protocol (Keyes *et al*, 2016). Female mice were used for sorting experiments at all time points and conditions to obtain maximal cell numbers. Single-cell suspensions were stained with antibodies at predetermined concentrations in a 100 µl staining buffer (PBS containing 5% FBS and 1% HEPES) per 10$^6$ cells. Stained cells were resuspended in DAPI in FACS buffer (Sigma) before analysis. Data were acquired on LSRII Analyzers (BD Biosciences) and then analyzed with FlowJo program. FACS was conducted using Aria Cell Sorters (BD Biosciences) into either staining buffer or TRIzol LS (Invitrogen).

## Histology

Skin tissue was fixed in PBS containing 10% formalin, paraffin embedded, sectioned (0.8 mm), and stained with hematoxylin and eosin by Histowiz Inc. Stained slides were scanned at 40× magnification using Aperio AT2. Slides were visualized, and epidermal thickness was analyzed manually based on morphological features of the epidermis using the Aperio Image Scope software. Each skin section was measured at 10 different locations at least 10 mm apart and averaged to obtain presented thickness value.

## RNA purification and quantitative PCR

Individual animals were used for qPCR experiments. Total RNA was purified from either whole skin biopsies, flash frozen, and then homogenized with a Bessman Tissue Pulverizer (SpectrumTM) or FACS-purified keratinocyte populations using Direct-zol RNA Mini-Prep Kit (Zymo Research) as per manufacturer's instructions. Equal amounts of RNA were reverse-transcribed using the superscript VILO cDNA Synthesis Kit (Invitrogen). cDNAs for each sample were normalized to equal amounts using primers against *Actb*. XpressRef Universal Total RNA (Qiagen) was used as a negative control to assess FACS population purity.

## Data availability

The primary read files as well as expression count files for the single-cell RNA sequencing datasets reported in this paper are available to download from GEO (accession number: GSE115765; http://www.ncbi.nlm.nih.gov/geo/query/acc.cgi?acc=GSE115765.

Expanded View for this article is available online.

## Acknowledgements

The authors would like to thank Thomas Boehm for his valuable support during the course of study and reading the manuscript; Christiane Happe and Ingrid Falk for their assistance with the mouse work and flow cytometry; Anja Nusser for critical reading of the manuscript; Sebastian Hobitz, Konrad Schuldes, Lukas Bischer, and Andreas Würch from the flow cytometry facility as well as Ulrike Bönisch, Chiara Bella and Laura Arrigoni from the deep sequencing facility of the Max Planck Institute of Immunobiology and Epigenetics. D.G. was supported by the Max Planck Society, the German Research Foundation (DFG) (SPP1937 GR4980/1-1, GR4980/3-1, and GRK2344 MeInBio), by the DFG under Germany's Excellence Strategy (CIBSS—EXC-2189—Project ID 390939984), by the European Research Council (ERC) (818846—ImmuNiche—ERC-2018-COG), and by the Behrens-Weise-Foundation.

## Author contributions

DG and S conceived the study and designed the experiments, S performed all FACS and scRNA-seq experiments as well as data analysis under the supervision of DG, MP performed thymocyte isolation from *Maf* KO mice and characterized the phenotype of *Maf* KO mice, JSH contributed to FACS and scRNA-seq experiments, PZ provided the dot plot figure code, MP and SN performed the psoriasis experiments, ES and UL contributed to *Sox13* and *Rorc* KO mice work, respectively, and MW, YT, and DRL supervised the *Sox13*, *Rorc*, and *Maf* KO studies, respectively. SS wrote the manuscript with guidance by DG and input from MP and DRL All authors edited the manuscript.

## Conflict of interest

The authors declare that they have no conflict of interest.

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
