## [Review Process File · The EMBO Journal]

Deciphering the Regulatory Landscape of Fetal and Adult gd T-cell Development by Single-cell RNA-sequencing

Dominic Gruen, Maria Pokrovskii, Josip Herman, Shruti Naik, Elisabeth Sock, Patrice Zeis, Ute Lausch, Michael Wegner, Yakup Tanriver, Dan Littman, and Sagar Sagar

DOI: [10.15252/embj.2019104159](https://doi.org/10.15252/embj.2019104159)

Corresponding author(s): Dominic Gruen (gruen@ie-freiburg.mpg.de)

Review Timeline:

Submission Date:	2nd Dec 19
Editorial Decision:	2nd Jan 20
Revision Received:	24th Mar 20
Editorial Decision:	9th Apr 20
Revision Received:	15th Apr 20
Accepted:	22nd Apr 20

Editor: Karin Dumstrei

Transaction Report:

Dear Dr. Gruen,

Thank you for submitting your manuscript for consideration by the EMBO Journal. It has now been seen by two referees whose comments are shown below.

As you can see, the referees appreciate the resource aspect of the analysis and is overall supportive of the work. However, they also both find that the manuscript is a difficult read and that this aspect along with others have to be addressed in order to consider publication here. Should you be able to address the concerns raised in full I would be interested in considering a revised version. Let me know if we need to discuss any points further - happy to do so

Thank you for the opportunity to consider your work for publication. I look forward to your revision.

with best wishes

Karin

Karin Dumstrei, PhD
Senior Editor
The EMBO Journal

When assembling figures, please refer to our figure preparation guideline in order to ensure proper formatting and readability in print as well as on screen:
<http://bit.ly/EMBOPressFigurePreparationGuideline>

Further information is available in our Guide For Authors:

The revision must be submitted online within 90 days; please click on the link below to submit the revision online before 1st Apr 2020.

Link Not Available

Referee #1:

The paper by Grun and colleagues is a data-packed description of RNA-seq profiling of murine $\gamma\delta$ T cells and their thymic precursors, in fetal and adult life. Although formally well presented and written, the density of findings and resource to overstatements make the paper a difficult read; furthermore, given the relative lack of interest or novelty of some data sets (see below), the paper should be amply revised to make it more appealing and accessible to the large audience of EMBO J.

This is my assessment of the 5 main claims of the paper:

- Identification of novel subtypes of immature and mature $\gamma\delta$ T cells:

It redefines previous subsets but the biological relevance of this resolution is somewhat unclear (see below) - may be however suitable for a resource-like paper;

- Identification of an unpolarized thymic population which expands in blood

While interesting, its biology remains unexplored. It thus comes across as an "unknown quantity", despite the authors' overstatement, which should be qualified.

- Infer continuous lineage trajectories

While bioinformatically sound, these lack validation in experimental precursor-product relationships, since the evolution of gene expression patterns may be less linear than expected; should be

discussed.

- Detailed comparison of fetal and adult $\gamma\delta$ T cell differentiation

Clearly one of the main points of the paper; found a surprising (based on the literature) and interesting level of similarity between fetal and adult $\gamma\delta$ T cell differentiation.

- Sequential activation of Sox13, Maf and Rorc in $\gamma\delta$ T17 cell commitment and their role in controlling TCR signaling strength. Interesting but lacking either novelty (see below) or mechanistic strength in the link to TCR signaling.

The paper clearly has two parts: the first is the descriptive and novel single-cell RNA-seq data; the second is a detailed exploration of the Sox13/ Maf/ Rorc network. The latter suffers from lack of novelty given the publication (online since 10th Dec 2018) by Zuberbuehler, Ciofani et al. Nat Immunol 2018, which dissects with great depth the Maf/ Rorc connection and impact on $\gamma\delta$ T17 cell specification. I therefore think the paper should be revised to focus on the novel and tune down (and show as supplementary data) the aspects that are redundant with this previous publication. Taking into account other comments below (including on Figs. 1 and 4), I would suggest the paper to be restructured into a shorter format with the (4-5 instead of 8) main figures focusing on the more novel and relevant aspects (current Figures 2, 3, 5 plus a selection of panels from Figs. 6 and 7).

Specific & technical issues:

Abstract

The claim to have established "their (Sox13, Maf, Rorc) role in controlling T cell receptor signaling strength" is an overstatement because the data provided are too thin and exclusively based on dysregulated expression of some putative TCR targets, which is open to alternative explanations. Namely, the impact of Sox13 deletion may affect a separate lineage of precursors, so that the net observations are simply an accumulation of signatures of the alternative pathway (namely, TCR-dependent and IFN γ -biased). Models of lineage competition or lineage diversion, rather than direct impact on TCR signaling in the same progenitors (in WT vs. Sox13^{-/-} mice), cannot be discriminated by the current experiments. This should be discussed and qualified.

Fig. S1: sorting strategy

E17.5 thymus:

- B: no lineage depletion except CD4 and CD8, complete fetal DN1 (Spidale et al., 2018 about DN1d?)

- D: fetal thymus is much different from thymi shown in B and C - older, dying?

6W thymus:

- F: lineage depletion, DN cells enriched by MACS

- G: pre- and post-selection $\gamma\delta$ T cells separation based on TCR levels. Does this also include CD24⁻ cells (fetal origin)?

- H: does the CD24⁺ immature population overlap with the CD25⁺ $\gamma\delta$ T cells?

Related to Fig. 1:

The authors employed:

4146 fetal cells : 30 clusters = 138 cells/cluster (from 24 embryos)

3235 adult cells : 24 clusters = 134 cells/cluster (from 11 mice)

How are these experimental differences taken into account?

It is unclear how $\gamma\delta$ T cells feature in Fig. 1 - although present in the figure panels (B-E) and in the title, the text focuses on the DN thymocyte heterogeneity. Should be clarified - and potentially shown as supplementary instead?

Also unclear what the reader can get from panels J-K?

DN cells (also Fig. S2):

Not excluding lineage+ cells from the sort of DN cells might include lineage+ cells especially in the adult.

Fetal: there are more clusters originating from DN1 cells (16,26,18...) which are not mentioned in the text but appear to be specific for fetal DN1 cells.

The analysis of the DN compartment seems to ignore an important paper - Spidale, Kang et al. Immunity 2018 - that showed that $\gamma\delta$ T17 cells originate from a discrete set of DN1d-e progenitors. It would be interesting to dissect the DN compartment with that level of resolution. Can the authors re-analyze their data by focusing on the respective markers? It would be important to know if their data supports the model by Kang and colleagues.

Related to Fig. 2:

- Adding maturation markers to better estimate if the cells are recent thymic emigrants, especially for the Ccr9+S1pr1+ subset?

- Mouse model e.g. Rag-GFP to estimate time since egress from thymus and potential maturation stages inside the blood pool which could explain relationships between clusters?

Related to Fig. 3:

The identification of an unpolarized thymic population which is highly represented in the blood is interesting, although its biology remains unexplored. Why have the authors not subjected this (sorted) population to in vitro differentiation assays, as to establish their functional potential?

Moreover, the conclusion that this population is "preferentially recruited from the thymus to the periphery" does not take into account an alternative: that unlike $\gamma\delta$ T17 and Il2rb+ cells, the unpolarized cells do not home to tissues (a known property of the other subsets) and thus relatively accumulates in the blood. Should be qualified.

Related to Fig. 4:

This figure is quite predictable lacks interest to be shown as main figure; I suggest presenting it as supplementary data.

Related to Figs. 6-7:

The redundancy of some pieces of data and their implications to the previous paper by Zuberbuehler, Ciofani et al. 2018 should be scrutinized and lead to a revised figure with the more novel aspects of the current study (with the rest being shown as supplementary). Namely, the thymic and peripheral $\gamma\delta$ T17 cell phenotypes of Maf-deficient and Rorc-deficient mice are well known.

Discussion

Zuberbuehler, Ciofani et al. 2018 have convincingly shown, through the use of various $\gamma\delta$ TCR transgenes, that differences in $\gamma\delta$ TCR signal strength result in graded expression of c-Maf, which directly controls the Rorc locus and thus $\gamma\delta$ T17 cell specification. How do the authors fit Sox13 into this model? This should be deeply discussed, trying to integrate both studies for the benefit of the

reader.

Other studies to be discussed in depth: Kernfeld et al. Immunity (also related to subsets that feature in Fig. 8); and Spidale, Kang et al. Immunity 2018 (as mentioned above).

Referee #2:

Summary:

In this work the authors comprehensively profile Gamma Delta T-cells in adult and fetal mouse thymus using single-cell RNA-sequencing, and study the activation of Sox13, Maf and Rorc during Gamma Delta T17 commitment using knockout mice.

While the data is substantial and comprehensive, and is likely to provide a useful resource for future studies, I find it, in some places, difficult to follow (see comment below). The analysis was done rigorously and most of the stages are explained in detail.

Major concerns:

1. The fact that you use thymus, blood and a peripheral tissues is an important aspect of this work, however the way the manuscript is written makes it less clear and very hard to follow: The blood and skin analyses are not directly mentioned in the abstract (there is one indirect mention of the blood) and are confusingly mentioned in the main text. The skin analysis should also be mentioned in the Introduction where you give the overall design. In results - the sentence "Vg5 is specifically expressed by skin-resident dendritic epidermal T cells (DETCs)." adds to this confusion - is this a known fact? (then this should have a proper reference) or do you see it in your skin data? (that has not yet been mentioned in the text). Please clarify this (and also structure the manuscript in an easier way to follow your different experiments).

2. Did you test the impact of freezing & thawing on the quality of the transcriptome and on possible biases in terms of gene expression? Similarly, is it known how FACS sorting procedure affects the transcriptome of these specific cells?

Minor concerns:

3. Parameters used for tSNE should be stated (and their choice should be justified)

4. Experimental design - was all of the mice sacrificed together or were there separate batches? How many litters of embryos have you used and do embryos from the same litter cluster differently than with embryos from other litters?

Non-essential suggestions for improving the study:

5. Can you decrease the size of each of the dots (representing a cell) in Fig 1 B-E? it's difficult to observe the structure of the tSNE and various clusters with this size of dots.

6. In Discussion, in:

"While we revealed an early upregulation

of the TCR recombination genes and overall enhanced proliferative capacity

in fetal versus adult thymocyte lineages, gene regulatory programs of gd T cell

differentiation and the emerging sub-types were surprisingly similar in fetal and adult thymi."

"Where" should be "were"

Response to Reviewer #1:

We thank Rev. 1 for his/her constructive criticism. Remarks of Rev. 1 are denoted in italics. Our responses to the reviewer are highlighted in blue.

The paper by Grun and colleagues is a data-packed description of RNA-seq profiling of murine $\gamma\delta$ T cells and their thymic precursors, in fetal and adult life. Although formally well presented and written, the density of findings and resource to overstatements make the paper a difficult read; furthermore, given the relative lack of interest or novelty of some data sets (see below), the paper should be amply revised to make it more appealing and accessible to the large audience of EMBO J.

We thank the referee for appreciating the density of the data and the depth of the analysis. Based on the comments of the referee, we have substantially revised the manuscript and provide answers to the comments below.

This is my assessment of the 5 main claims of the paper:

*- Identification of novel subtypes of immature and mature $\gamma\delta$ T cells:
It redefines previous subsets but the biological relevance of this resolution is somewhat unclear (see below) - may be however suitable for a resource-like paper;*

Given the depth and amount of data already included in the analysis, exploring the biological relevance of all the subsets would require a substantial amount of experimental follow-up analysis and will be out of the scope of the current manuscript. We have done additional experiments to address the comments regarding the unpolarized thymic subpopulation (see below), providing experimental validation of one of the novel sub-populations. Although these experiments helped to better characterize this population, we agree that the major strength of our work is the resource aspect, and we therefore have resubmitted the revised version as a resource article.

*- Identification of an unpolarized thymic population which expands in blood
While interesting, its biology remains unexplored. It thus comes across as an "unknown quantity", despite the authors' overstatement, which should be qualified.*

In order to further explore the biology of this subpopulation, we used antibodies against CCR9 and S1PR1 to isolate and perform further experiments. Unfortunately, single-cell RNA sequencing of FACS-sorted CCR9⁺ S1PR1⁺ $\gamma\delta$ thymocytes revealed that these two markers cannot be used for the enrichment of this subpopulation (most likely due to the discrepancy between RNA and protein levels) thereby preventing us to perform in vitro differentiation or activation assays on these cells from the thymus. However, we were able to enrich this subpopulation from the peripheral blood using CD44. The CD44^{neg} gate specifically contained the Ccr9⁺ S1pr1⁺ $\gamma\delta$ subset (Revised figure 3). Stimulation of these cells with PMA/ionomycin revealed that they produce high levels of TNF- α , IL-2 and IFN- γ (see below for the further details). We further demonstrate a correspondence of the thymic Ccr9⁺ S1pr1⁺ population to the Ccr9⁺ S1pr1⁺ populations from the peripheral blood and lymph nodes by integration of single-cell RNA-seq data.

*- Infer continuous lineage trajectories
While bioinformatically sound, these lack validation in experimental precursor-product relationships, since the evolution of gene expression patterns may be less linear than expected; should be discussed.*

We agree with the referee and acknowledge this general problem in reconstructing lineage trees using scRNA-seq data without lineage tracing experiments. However, in this manuscript, we have used extensive previous literature combined with the StemID algorithm to derive the differentiation trajectories. For example, FLT3⁺ DN1 cells are defined as the root as they are considered developmentally early ETPs. Similarly, the previously known chronological order of analyzed sub-populations, i.e., CD25⁺ $\gamma\delta$ T cell progenitors, immature CD24⁺ $\gamma\delta$ T cells and mature CD24^{neg} $\gamma\delta$ T cells, was reflected by our derived trajectories. Furthermore, special care has been taken regarding the clusters which do not fall on the continuous differentiation manifold such as mature adult $\gamma\delta$ T17 cluster 10. Based on the expression of *Ccr6*, this cluster seems to be non-ETP derived and may arise from Sox13-ECFP⁺ DN1d progenitors described by Kang and colleagues, and therefore was excluded from the trajectory analysis. This is discussed in detail in the revised discussion section. Furthermore, some of the descriptive data derived from the pseudo-temporal analysis have now been moved to an extended figure (Figure EV6).

- Detailed comparison of fetal and adult $\gamma\delta$ T cell differentiation

Clearly one of the main points of the paper; found a surprising (based on the literature) and interesting level of similarity between fetal and adult $\gamma\delta$ T cell differentiation.

We thank the referee for acknowledging our extensive analysis of fetal and adult $\gamma\delta$ T cell differentiation revealing that the same gene networks operate at both time points. To further elucidate these similarities in the revised manuscript we have performed additional analysis, e.g., integration of the fetal and adult datasets to identify the common and unique subtypes at these two stages.

- Sequential activation of Sox13, Maf and Rorc in $\gamma\delta$ T17 cell commitment and their role in controlling TCR signaling strength. Interesting but lacking either novelty (see below) or mechanistic strength in the link to TCR signaling.

We have addressed this issue by removing redundant data, e.g., showing that *Maf* KO mice lack $\gamma\delta$ T17 cells in the small and large intestinal lamina propria. Moreover, to focus on the novel aspects, we compressed the representation of the KO datasets into one main figure 5 and moved the rest to extended figures 8 and 9 (see below for details).

The paper clearly has two parts: the first is the descriptive and novel single-cell RNA-seq data; the second is a detailed exploration of the Sox13/ Maf/ Rorc network. The latter suffers from lack of novelty given the publication (online since 10th Dec 2018) by Zuberbuehler, Ciofani et al. Nat Immunol 2018, which dissects with great depth the Maf/ Rorc connection and impact on $\gamma\delta$ T17 cell specification. I therefore think the paper should be revised to focus on the novel and tune down (and show as supplementary data) the aspects that are redundant with this previous publication. Taking into account other comments below (including on Figs. 1 and 4), I would suggest the paper to be restructured into a shorter format with the (4-5 instead of 8) main figures focusing on the more novel and relevant aspects (current Figures 2, 3, 5 plus a selection of panels from Figs. 6 and 7).

Following the referee's recommendation, we have shortened the revised manuscript to 5 main data figures plus 1 summary figure. Redundant data have been removed or moved to the extended version figures. On the other hand, novel data regarding the biology of *Ccr9*⁺*S1pr1*⁺ $\gamma\delta$ subset have been added in the revised Figure 3 (details

provided below).

Specific & technical issues:

Abstract

The claim to have established "their (Sox13, Maf, Rorc) role in controlling T cell receptor signaling strength" is an overstatement because the data provided are too thin and exclusively based on dysregulated expression of some putative TCR targets, which is open to alternative explanations. Namely, the impact of Sox13 deletion may affect a separate lineage of precursors, so that the net observations are simply an accumulation of signatures of the alternative pathway (namely, TCR-dependent and IFN γ -biased). Models of lineage competition or lineage diversion, rather than direct impact on TCR signaling in the same progenitors (in WT vs. Sox13^{-/-} mice), cannot be discriminated by the current experiments. This should be discussed and qualified.

The sentence has been removed from the abstract. However, we think that the impact of Sox13 and Maf deletion on TCR signaling still holds true. The conclusion that Sox13 and Maf-deleted cells exhibit higher expression of TCR signaling related genes is not based on few genes but also on the gene set enrichment analysis. Nevertheless, we agree with the referee that the current experiments do not allow us to exclude the alternate explanations. Therefore, we have toned down the claims related to the role of Sox13 and Maf in regulating the TCR signaling and have also addressed this in the discussion of the revised manuscript.

Fig. S1: sorting strategy

E17.5 thymus:

- B: no lineage depletion except CD4 and CD8, complete fetal DN1 (Spidale et al., 2018 about DN1d?)

In this analysis, we focused on ETPs, i.e., c-KIT⁺CD44⁺CD25^{neg} DN1 progenitors. For clarity, this information has been updated in the text and DN1 cells are now termed c-KIT⁺ DN1s.

- D: fetal thymus is much different from thymi shown in B and C - older, dying?

The thymus in D is indeed older. Our timed mating had some variability (+1 day) due to which some embryos were older than E17.5. In the text and figures, we now define the stage of fetal mouse thymi as E17.5-E18.5. Furthermore, in all experiments, dead cells were excluded using DAPI (see methods section).

6W thymus:

- F: lineage depletion, DN cells enriched by MACS

The details have been mentioned in the methods under the section "Magnetic enrichment of adult double negative thymocytes".

- G: pre- and post-selection $\gamma\delta$ T cells separation based on TCR levels. Does this also include CD24⁻ cells (fetal origin)?

Pre- and post-selected $\gamma\delta$ T cells (very rare in the adult thymus) were sorted based on CD25 expression. We did not find CD24^{neg} $\gamma\delta$ T cells in this compartment.

- H: does the CD24⁺ immature population overlap with the CD25⁺ $\gamma\delta$ T cells?

Indeed, few cells from CD24⁺ immature population co-clustered with CD25⁺ $\gamma\delta$ T cell clusters. For clarity, we have now included bar plots of cell type composition in the revised Figure 1 (D and G).

Related to Fig. 1:

The authors employed:

4146 fetal cells : 30 clusters = 138 cells/cluster (from 24 embryos)

3235 adult cells : 24 clusters = 134 cells/cluster (from 11 mice)

How are these experimental differences taken into account?

Our general approach to decide on appropriate number of cells is to first sequence few plates from different sorting gates and perform the analysis to characterize the heterogeneity in the dataset. If clustering analysis reveals substantial heterogeneity, we further sequence more plates to increase the cell numbers in different clusters sufficiently in order to resolve this heterogeneity. Using this strategy, we ended up sequencing the above mentioned number of cells. We believe that sequencing more cells would just lead to an increase in the number of cells per cluster rather than the discovery of additional populations, i.e., to an increase in cluster numbers. Therefore, we believe that the size of the datasets is sufficient and appropriate to resolve the heterogeneity at the respective stage. Importantly, we do not claim that cluster sizes reflect the frequency of the respective population in the thymus.

It is unclear how $\gamma\delta$ T cells feature in Fig. 1 - although present in the figure panels (B-E) and in the title, the text focuses on the DN thymocyte heterogeneity. Should be clarified - and potentially shown as supplementary instead?

Also unclear what the reader can get from panels J-K?

Since we were interested in understanding the development of $\gamma\delta$ T cells and deriving the differentiation trajectories starting from the earliest progenitors, we also sequenced double negative thymocytes capable of giving rise to $\gamma\delta$ T cells. Furthermore, we were also interested in (computationally) investigating if early ETPs or DN2 progenitors exhibit an early $\gamma\delta$ T cell bias. Our results did not provide evidence for such an early cell fate bias on the transcriptome level. Since we feel that this finding as well as the elucidation of DN thymocyte heterogeneity in general is of interest to the community, and our differentiation trajectories included in the dataset are also derived from the early ETPs, we decided to keep these data in the manuscript. However, as suggested by the referee, we revised Figure 1 and removed the non-essential aspects such as previous Figure panels 1J and K.

DN cells (also Fig. S2):

Not excluding lineage+ cells from the sort of DN cells might include lineage+ cells especially in the adult.

We agree with this point. However, we intentionally avoided the use of any lineage cocktail while sorting as we aimed to be as unbiased as possible. Moreover, our preliminary data showed that except for the c-KIT⁺ DN1 gate (from which we also captured few $\gamma\delta$ T cells), other DN (DN2 and DN3) gates are relatively pure and we do not get cells of other lineages. Also we note that the single-cell resolution allows us to detect and remove these lineage+ cells from the downstream data analysis, if necessary.

Fetal: there are more clusters originating from DN1 cells (16,26,18...) which are not mentioned in the text but appear to be specific for fetal DN1 cells.

Indeed, as mentioned above, both in the fetal and the adult data we found outlier clusters comprising few cells, including $\gamma\delta$ T cells from the c-KIT⁺ DN1 gate. Because of the space limitations and to avoid confusion amid the already extensive analysis,

we focused our attention on the clusters with more than 15 cells. To annotate all clusters comprehensively, we added to the revised manuscript a table of differentially upregulated genes in each clusters enabling further investigation based on these marker genes.

The analysis of the DN compartment seems to ignore an important paper - Spidale, Kang et al. Immunity 2018 - that showed that $\gamma\delta T17$ cells originate from a discrete set of DN1d-e progenitors. It would interesting to dissect the DN compartment with that level of resolution. Can the authors re-analyze their data by focusing on the respective markers? It would be important to know if their data supports the model by Kang and colleagues.

We apologize for not citing and discussing the important paper by Kang and colleagues. In this study, we only sorted c-KIT⁺ DN1 progenitors capable of giving rise to both $\alpha\beta$ and $\gamma\delta$ T cells. Therefore, we capture only DN1a and DN1b cells in our dataset and lack DN1c-e progenitors. The study by Kang and colleagues was published when this manuscript was already approaching the final stage (November 2018). However, as the referee has suggested, we have discussed this in detail in our revised version (see below).

Related to Fig. 2:

- Adding maturation markers to better estimate if the cells are recent thymic emigrants, especially for the Ccr9+S1pr1+ subset?
- Mouse model e.g. Rag-GFP to estimate time since egress from thymus and potential maturation stages inside the blood pool which could explain relationships between clusters?

In the adult thymus, the Ccr9⁺S1pr1⁺ subset is indeed coming from the immature $\gamma\delta$ T cell gate, i.e., they are CD24⁺. In the revised version, we have shown that almost all the CD44^{neg} $\gamma\delta$ T cells in the blood lack lineage-specifying marker genes and are Ccr9⁺S1pr1⁺, which supports our argument that these cells are naïve $\gamma\delta$ T cells. Transcriptionally, the blood Ccr9⁺S1pr1⁺ cells also express Sell (encoding CD62L, data not shown) which further supports the idea that these cells are naïve $\gamma\delta$ T cells and correspond to CD44^{lo} CD62L^{hi} adaptive-like $\gamma\delta$ T cells.

Related to Fig. 3:

The identification of an unpolarized thymic population which is highly represented in the blood is interesting, although its biology remains unexplored. Why have the authors not subjected this (sorted) population to in vitro differentiation assays, as to establish their functional potential?

Moreover, the conclusion that this population is "preferentially recruited from the thymus to the periphery" does not take into account an alternative: that unlike $\gamma\delta T17$ and Il2rb+ cells, the unpolarized cells do not home to tissues (a known property of the other subsets) and thus relatively accumulates in the blood. Should be qualified.

Since we were not able to enrich this subset from the thymus using antibodies against S1PR1 and CCR9 (as determined by scRNA-seq of samples sorted accordingly), we could not sort and perform in vitro differentiation or activation assays. Since the referee has raised concerns regarding the naïve status of this population, we checked the expression of Cd44 and other naïve marker genes (discussed above) and found that these cells express low to no Cd44 transcripts (revised Figure 3C). Therefore, we hypothesized that we could use CD44 as a marker to sort these cells from the blood and perform activation assays to access

their cytokine profile. Indeed, we were able to enrich this population in the CD44^{neg} gate (revised Figure 3). Unfortunately, the limited amount of material did not permit to perform antibody staining (we recovered only 6000 Cd44^{neg} cells from the pooled blood of 3-4 mice to perform stimulation assays) and therefore performed scRNA-seq after stimulating them with PMA/ionomycin. Our results indicate that these cells produce *Tnf*, *Il2* and *Ifng* after stimulation.

In order to answer referee's concern that these cells may not home to tissues and therefore accumulate in the blood, we performed scRNA-seq of lymph node $\gamma\delta$ T cells, and, surprisingly, found an even bigger fraction of the *Ccr9*⁺*S1pr1*⁺ subset in lymph nodes, which argues against the hypothesis that they do not migrate to the tissues. However, it will be interesting to investigate in the future if they selectively migrate to blood and secondary lymphoid organs but not to the epithelial tissues. Our preliminary scRNA-seq data from other epithelial tissues reveals that these cells are also present in the liver albeit at lower frequency (data not shown).

Importantly, our dataset integration shown in Figure 3M indicates that the *Ccr9*⁺*S1pr1*⁺ marks corresponding sub-populations in thymus, peripheral blood, and lymph nodes.

Related to Fig. 4:

This figure is quite predictable lacks interest to be shown as main figure; I suggest presenting it as supplementary data.

The figure has been moved to the extended version.

Related to Figs. 6-7:

*The redundancy of some pieces of data and their implications to the previous paper by Zuberbuehler, Ciofani et al. 2018 should be scrutinized and lead to a revised figure with the more novel aspects of the current study (with the rest being shown as supplementary). Namely, the thymic and peripheral $\gamma\delta$ T17 cell phenotypes of *Maf*-deficient and *Rorc*-deficient mice are well known.*

We agree with the referee regarding the redundancy of several *Maf* KO datasets presented in the main figure of our manuscript. Following the advice, we have removed the known aspects of the *Maf* KO mice from the manuscript such as the absence of RORgt⁺ cells from the small and large intestinal lamina propria. Furthermore, as the referee suggested, we have now merged the data from *Sox13* KO and *Maf* KO mice into one main figure (Figure 5). Since the psoriasis phenotype of the *Maf* KO mice is not reported before, this has been kept in the main figure 5. The scRNA-seq dataset of *Rorc* KO mice is included in the supplement as this is essential in claiming that the expression of *Sox13* remains unaffected in the absence of *Rorc*, indicating the lack of a feedback loop between *Sox13* and *Rorc*.

Discussion

*Zuberbuehler, Ciofani et al. 2018 have convincingly shown, through the use of various $\gamma\delta$ TCR transgenes, that differences in $\gamma\delta$ TCR signal strength result in graded expression of *c-Maf*, which directly controls the *Rorc* locus and thus $\gamma\delta$ T17 cell specification. How do the authors fit *Sox13* into this model? This should be deeply discussed, trying to integrate both studies for the benefit of the reader.*

We thank the referee for these constructive comments. We have significantly revised the discussion section of the manuscript. We would like to point out that our results go beyond the finding of Ciofani and colleagues that modulating the levels of TCR results in inversely correlated expression levels of *Maf*. We observe that *Sox13* and

Maf deleted cells upregulate or cannot downregulate the TCR signaling related genes. This observation is not only based on few marker genes such as Nr4a1/Cd69/Cd5 etc., but also supported by gene set enrichment analysis revealing that KO cells exhibit higher expression of gene sets related to TCR, PI3/AKT and MAPK signaling pathways. Although our experimental setup did not allow us to conclusively prove the role of these two transcription factors in modulating TCR signaling, we noticed a similar observation made by Ciofani and colleagues as they also see an upregulation of TCR signaling related gene sets in their KO cells. Furthermore, we would like to emphasize that the transgenic TCR signaling experiment performed by Ciofani and colleagues was done in a completely different experimental set-up (using *Rag*-deficient DN progenitors), and, hence, could not address the role of *Maf* in regulating TCR signaling, but rather revealed how TCR signaling affects *Maf* expression in their system. Due to the space restrictions, we could not discuss this issue extensively, but we have included a detailed paragraph in the discussion regarding the results related to TCR signaling and how *Sox13* might work together with *Maf* in the process of $\gamma\delta$ T17 specification.

Other studies to be discussed in depth: Kernfeld et al. Immunity (also related to subsets that feature in Fig. 8); and Spidale, Kang et al. Immunity 2018 (as mentioned above).

We again thank the referee for recommending to add the important findings of these studies, which we have now included in the revised discussion.

Response to Reviewer #2:

We thank Rev. 2 for his/her insightful comments. Remarks of Rev. 1 are denoted in italics. Our responses to the reviewer are highlighted in blue.

Summary:

In this work the authors comprehensively profile Gamma Delta T-cells in adult and fetal mouse thymus using single-cell RNA-sequencing, and study the activation of Sox13, Maf and Rorc during Gamma Delta T17 commitment using knockout mice. While the data is substantial and comprehensive, and is likely to provide a useful resource for future studies, I find it, in some place, difficult to follow (see comment below). The analysis was done rigorously and most of the stages are explained in detail.

We thank the referee for appreciating the amount of data and depth of the analyses in our manuscript. We agree that this work may provide a useful resource to help the community to interpret their data in terms of cell subtypes as well as to design future studies. Considering the comments and recommendations of both referees, we have now substantially revised the manuscript to compress its size and make it more focused. Furthermore, we have now submitted the manuscript as a resource article. Below we provide a point by point response to the referee's concerns.

Major concerns:

1. The fact that you use thymus, blood and a peripheral tissues is an important aspect of this work, however the way the manuscript is written makes it less clear and very hard to follow: The blood and skin analyses are not directly mentioned in the abstract (there is one indirect mention of the blood) and are confusingly mentioned in the main text. The skin analysis should also be mentioned in the Introduction where you give the overall design. In results - the sentence "Vg5 is specifically expressed by skin-resident dendritic epidermal T cells (DETCs)." adds to this confusion - is this a known fact? (then this should have a proper reference) or do you see it in your skin data? (that has not yet been mentioned in the text). Please clarify this (and also structure the manuscript in an easier way to follow your different experiments).

We are sorry for the confusion regarding the tissues used for scRNA-seq. We have revised the manuscript to make this clearer. In the original version of the manuscript, we have only used two different tissues – thymus and the blood. No other tissue was sequenced including the skin. As mentioned in the introduction, $\gamma\delta$ T cells expressing different variable TCR chains during development localize to different tissues. To understand their transcriptional signature during their development in the thymus, we sequenced $\gamma\delta$ T cells expressing different variable chains from the fetal and adult thymus including Vg5 $\gamma\delta$ T cells which migrate to the skin. We have not profiled them directly from the skin. It is indeed known that Vg5 cells reside in the skin. As suggested by the other referee, the figure has been moved to supplement and the text has been reduced to avoid the confusion. During the revision, lymph node $\gamma\delta$ T cells have been sequenced in addition to cells from thymus and blood in order to investigate whether the $Ccr9^+S1pr1^+$ subset is also expanded in the secondary lymphoid organs. We hope that the revised version is clearer and easier to read compared to the previous version.

2. Did you test the impact of freezing & thawing on the quality of the transcriptome and on possible biases in terms of gene expression? Similarly, is it known how FACS

sorting procedure affects the transcriptome of these specific cells?

We thank the reviewer for raising this concern. Indeed, we were also concerned that freezing and thawing may lead to changes in the transcriptome or death of specific thymocyte subsets. Therefore, we performed a combined analysis of $\gamma\delta$ T cells sorted from WT mice from our facility (comprised of only fresh thymocytes) with the cells sorted from *Maf*^{fl/fl} mice (frozen), which served as littermate control for the *Maf* KO mice. The fresh as well as frozen cells intermingled very well in a dimensional reduction representation of the data, and we did not see any technical batch effect or effect of freezing on transcriptome during the data analysis. One example is shown below where frozen $\gamma\delta$ T cells from *Maf*^{fl/fl} and fresh WT B6 from our facility (depicted in green and red color, respectively) were well mixed:

Moreover, special care has been taken to include proper controls. For instance, frozen KO cells were always compared with frozen WT littermate controls from the same facility. Of note, *Sox13*, *Maf* and *Rorc* KO cells came from three different labs/facilities and frozen littermate controls from the respective lab were always used for differential gene expression and GSEA analysis. Moreover, DAPI staining was always performed to label and exclude dead cells from sorting. Extreme care has been taken while isolating the cells which were always kept cool on ice. As far as the concern of transcriptome changes due to FACS is concerned, we would like to emphasize that many of $\gamma\delta$ T cell subsets are very rare (especially in the adult thymus) and, therefore, FACS is the only option to profile these cells. Special care has been taken to perturb them as little as possible. For example, they were sorted at low flow rate/pressure, and plates and samples were always cooled during the sorting procedure to minimize potential effects on in the transcriptome.

Minor concerns:

3. *Parameters used for tSNE should be stated (and their choice should be justified)*

We agree with the reviewer that the choice of perplexity parameter is indeed important to control the locality of the distribution. Our general practice is to run tSNE with different perplexity parameters and to choose a value which is not too small and not too large to avoid any artefacts in the low dimensional manifold. Around this value the tSNE representation has to be relatively stable upon changes in the perplexity. Below we provide some examples of the fetal and adult data where dimensionality reduction was performed using different values of the perplexity parameter. Overall, the structure of the data remained fairly stable across different values and the default value (set to 30) of the RaceID3 algorithm was used to represent all the datasets in the manuscript.

4. *Experimental design - was all of the mice sacrificed together or were there separate batches? How many litters of embryos have you used and do embryos from the same litter cluster differently than with embryos from other litters?*

For all experiments from the thymi except fetal *Rorc* KO mice, mice were sacrificed in a minimum of two batches (two independent experiments). We have now updated the figure legends to provide this information. For each independent experiment, all embryos (3-8) from one pregnant female were pooled. As mentioned in the text, we did not see any batch associated variability. All data from each genotype or age intermingled well across all biological replicates irrespective of the day of the experiment or litters.

Non-essential suggestions for improving the study:

5. *Can you decrease the size of each of the dots (representing a cell) in Fig 1 B-E? it's difficult to observe the structure of the tSNE and various clusters with this size of dots.*

We have reduced the dot size of the tSNE maps in revised figures 1 and 2.

6. *In Discussion, in:*

"While we revealed an early upregulation of the TCR recombination genes and overall enhanced proliferative capacity in fetal versus adult thymocyte lineages, gene regulatory programs of gd T cell differentiation and the emerging sub-types were surprisingly similar in fetal and adult thymi."

"Where" should be "were"

Based on the comments of the referee 1, we have significantly changed the discussion and this sentence has been modified.

Dear Dr. Gruen,

Thank you for submitting your revised manuscript to The EMBO Journal. Your study has now been re-reviewed by the referees and their comments are provided below.

As you can see from the comments, both referees support publication in The EMBO Journal. They raise a number of good points that I would like to ask you address in a revised version. No new experiments are need - just some clarifications in the text (ref #1 and 2) and adding data from the point-by-point response into the main MS file (referee #2).

When you submit the revised manuscript will you also please take care of the following points:

- You have at the moment 9 EV figures, but can only have 5. You can the extra figures to the appendix. Please note the appendix should have a ToC. Please also see our author guidelines
- Keywords are missing
- Table EV1 should have a legend - please add it as a separate tab.
- Figure 5N is missing scale bars
- Please re-label Data and materials availability as Data Availability Section
- The Materials & Methods section needs moving to before the Acknowledgements.
- I have asked our publisher to do their pre-publication check on this manuscript but have not received their comments yet. I will pass them on as soon as I get them.
- We include a synopsis of the paper (see <http://emboj.embopress.org/>). Please provide me with a general summary statement and 3-5 bullet points that capture the key findings of the paper.
- We also need a summary figure for the synopsis. The size should be 550 wide by 400 high (pixels).

You can use the link below to upload the revised manuscript.

That should be all! Let me know if we need to discuss anything further.

Congratulations on a nice study

With best wishes

Karin

Karin Dumstrei, PhD
Senior Editor
The EMBO Journal

- a point-by-point response to the referees' comments, with a detailed description of the changes made (as a word file).

- a word file of the manuscript text.

- individual production quality figure files (one file per figure)

- a complete author checklist, which you can download from our author guidelines (<https://www.embopress.org/page/journal/14602075/authorguide>).

- Expanded View files (replacing Supplementary Information)

Further information is available in our Guide For Authors:

The revision must be submitted online within 90 days; please click on the link below to submit the revision online before 8th Jul 2020.

Link Not Available

Referee #1:

The authors have greatly improved the paper by re-organizing figures, clarifying text and providing some new data. In this regard, there is only one minor aspect to clarify in the text, which is the apparent paradox that the novel "unpolarised" gd T cell population (present in the thymus and blood) they describe makes high levels of IFN-g upon just PMA/ ionomycin stimulation, without the need for any polarising cytokines over time - it thus seems these cells have a type 1 effector default, but not yet fully mature, requiring peripheral signals (in an adaptive fashion). A small comment should be added to the final text.

Referee #2:

The authors have addressed my concerns and the manuscript is more clearly written.

I would suggest that the two analyses that the authors added in response to my second and third question will be added as supporting information, along with a description of the analysis parameters (i.e. which computational procedure was done, what is shown in each axis, etc)

Response to Reviewer #1:

We thank Rev. 1 for his/her constructive criticism. Remarks of Rev. 1 are denoted in italics. Our responses to the reviewer are highlighted in blue.

The authors have greatly improved the paper by re-organizing figures, clarifying text and providing some new data. In this regard, there is only one minor aspect to clarify in the text, which is the apparent paradox that the novel "unpolarised" $\gamma\delta$ T cell population (present in the thymus and blood) they describe makes high levels of IFN- γ upon just PMA/ ionomycin stimulation, without the need for any polarising cytokines over time - it thus seems these cells have a type 1 effector default, but not yet fully mature, requiring peripheral signals (in an adaptive fashion). A small comment should be added to the final text.

We thank the referee for appreciating the revised version of the manuscript. Following referee's advice we have added the following sentence in the results section ' $Ccr9^+$ $S1pr1^+$ $\gamma\delta$ T cells represent a major subset of blood and lymph node $\gamma\delta$ T cells and produce IFN- γ , TNF- α and IL-2 upon stimulation':

"Our results suggest that $Ccr9^+$ $S1pr1^+$ $\gamma\delta$ T cells are a subset of IFN- γ producing $\gamma\delta$ T cells but exit the thymus in an immature state and are polarized in the periphery in an adaptive-like fashion."

Also, we added the following sentence at the end of the second paragraph of the discussion: "Therefore, although these cells leave the thymus in a functionally immature state, they are already primed to produce IFN- γ and are likely to acquire their effector phenotype in the periphery through an adaptive-like mechanism."

Response to Reviewer #2:

We thank Rev. 2 for his/her constructive criticism. Remarks of Rev. 2 are denoted in italics. Our responses to the reviewer are highlighted in blue.

The authors have addressed my concerns and the manuscript is more clearly written.

We thank the referee for appreciating the revised version of the manuscript.

I would suggest that the two analyses that the authors added in response to my second and third question will be added as supporting information, along with a description of the analysis parameters (i.e. which computational procedure was done, what is shown in each axis, etc)

The data regarding the perplexity of t-SNE is now added as an Appendix Figure S5 in the revised manuscript. Parameters used for t-SNE are now included in the methods section. Furthermore, we have now done a more thorough analysis of fresh cells with frozen cells at both fetal and adult time points and included the data in Appendix Figure S6. Moreover, a new section in the Methods – ‘Combined analysis of $\gamma\delta$ T cells from fresh and frozen thymocytes’ is added to the manuscript providing the details of the parameters used to do this comparative analysis.

Dear Dominic,

Thank you for submitting your revised manuscript to The EMBO Journal. I have now had a chance to take a careful look at everything and all looks good. I am therefore very pleased to accept the manuscript for publication here.

Congratulations on a super nice study!

With best wishes

Karin

Karin Dumstrei, PhD
Senior Editor
The EMBO Journal

Please note that it is EMBO Journal policy for the transcript of the editorial process (containing referee reports and your response letter) to be published as an online supplement to each paper. If you do NOT want this, you will need to inform the Editorial Office via email immediately. More information is available here: http://emboj.embopress.org/about#Transparent_Process

Your manuscript will be processed for publication in the journal by EMBO Press. Manuscripts in the PDF and electronic editions of The EMBO Journal will be copy edited, and you will be provided with page proofs prior to publication. Please note that supplementary information is not included in the proofs.

Should you be planning a Press Release on your article, please get in contact with embojournal@wiley.com as early as possible, in order to coordinate publication and release dates.

If you have any questions, please do not hesitate to call or email the Editorial Office. Thank you for your contribution to The EMBO Journal.

EMBO Press encourages all authors and reviewers to associate an Open Researcher and Contributor Identifier (ORCID) to their account. ORCID is a community-based initiative that provides an open, non-proprietary and transparent registry of unique identifiers to help disambiguate research contributions.

Currently, our records indicate that the ORCID for your account is 0000-0002-3364-5898.

Please click the link below to modify this ORCID:
Link Not Available

YOU MUST COMPLETE ALL CELLS WITH A PINK BACKGROUND ↓
PLEASE NOTE THAT THIS CHECKLIST WILL BE PUBLISHED ALONGSIDE YOUR PAPER

Corresponding Author Name: Dominic Gruen
Journal Submitted to: The EMBO Journal
Manuscript Number: EMBOJ-2019-104159